# Ca-dimers, solvent layering, and dominant electrochemically active species in Ca(BH$_4$)$_2$ in THF

Ana Sanz Matias [1,2], Fabrice Roncoroni [1,2], Siddharth Sundararaman [1,2] & David Prendergast [1,2] ✉

Divalent ions (Mg, Ca, and Zn) are being considered as competitive, safe, and earth-abundant alternatives to Li-ion electrochemistry, but present challenges for stable cycling due to undesirable interfacial phenomena. We explore the formation of electroactive species in the electrolyte Ca(BH$_4$)$_2$|THF using molecular dynamics coupled with a continuum model of bulk and interfacial speciation. Free-energy analysis and unsupervised learning indicate a majority population of neutral Ca dimers and monomers with diverse molecular conformations and an order of magnitude lower concentration of the primary electroactive charged species – the monocation, CaBH$_4^+$ – produced via disproportionation of neutral complexes. Dense layering of THF molecules within ~1 nm of the electrode surface strongly modulates local electrolyte species populations. A dramatic increase in monocation population in this interfacial zone is induced at negative bias. We see no evidence for electrochemical activity of fully-solvated Ca$^{2+}$. The consequences for performance are discussed in light of this molecular-scale insight.

To increase the rate of conversion to renewable energy sources, electrification of various energy-intensive aspects of society is underway. The concomitant demand for electrochemical energy storage solutions increasingly highlights the limits of Li-ion technologies with respect to performance, safety, and sustainability. Multivalent ions, such as Mg$^{2+}$, Ca$^{2+}$, Zn$^{2+}$, or even Al$^{3+}$, offer more earth-abundant alternatives, some with higher theoretical specific capacity and reduced safety concerns due to self-passivation of metal anodes[1–5].

However, realizing performant electrochemical cells using these ions is hindered due to various issues driven by interfacial phenomena: low power output and charging rates due to large overpotentials and associated electrolyte decomposition and interphase growth[6–10]. At issue is our lack of understanding of the specific complex nature of solvation in suitable electrolytes for multivalent ions and the identification of which of these species are active at the electrode-electrolyte interface and why[11]. We may be biased by familiarity with aqueous solutions and the ability of water to generate perfect electrolytes, with fully dissociated and solvated ions, for many salts. However, organic solvents (required for a sufficiently wide window of electrochemical stability in batteries), with typically lower dielectric constants and larger molecular sizes, have increased residence times for coordinating highly charged cations and have relatively little interaction with anions, unlike ambipolar water molecules which more easily solvate both charges.

Molecular dynamics provides a window to the inner workings of electrolytes, revealing details of the coordination of cations by solvent molecules and anions[12,13]. However, in the study of highly-charged species in poor dielectrics, we must take care to avoid sampling only a limited set of coordination states due to their long lifetimes and unavoidable limitations in computing time and complexity. Free-energy sampling allows us the opportunity to pose fundamental questions regarding the chemical composition of a poor electrolyte and the mechanisms by which its solvated species interconvert[14–19]. Mining the large quantities of compositional and conformational data produced

---

[1]Joint Center for Energy Storage Research, Lawrence Berkeley National Laboratory, Berkeley, CA 94720, USA. [2]The Molecular Foundry, Lawrence Berkeley National Laboratory, Berkeley, CA 94720, USA. ✉e-mail: dgprendergast@lbl.gov

by these simulations presents its own challenge. Here, we rely on recently developed unsupervised learning approaches[20] to provide a faster path to extracting molecular-scale insight and guidance for future experiments to validate our predictions.

Calcium is a competitive multivalent candidate largely because its redox potential is just 0.2 V above Li (lower than Mg, Zn, or Al). Despite decades of research[21], only in 2016 was reversible plating and stripping on calcium metal achieved[22]. Room-temperature reversibility followed swiftly in 2018, using Ca/$BH_4^-$/THF[23]. Hydride-terminated anions such as $BH_4^-$ offer high efficiency without significant passivation[23] partly due to solvent and salt stability against the anode[24] in comparison with typical Li-ion salts and solvents[25]. Newer, more complex, hydride-borane-based electrolytes, such as carba-closoborane in mixed solvents (DME/THF)[26], with similar performance and reversibility, exhibit a wider electrochemical stability window and have recently shown good long-term operation characteristics[27]. This promise makes it even more important to understand the key aspects of hydride-based[28] anion electrolytes. Hence, we study $Ca(BH_4)_2$ in THF as a promising electrolyte candidate for room-temperature, reversible calcium deposition with best-in-class efficiency and minimum parasitic reactions—with complex bulk and largely unknown interfacial solvation environments[6,23,24,29–35].

We can also contrast its behavior with $Mg(TFSI)_2$ in THF, studied recently using free energy sampling[36]. Recent investigations of the complexation in the bulk electrolyte[29,37] detected Ca-dimers ($Ca_2(BH_4)_4$) experimentally. Dimers were suggested to facilitate the disproportionation of neutral monomers into anions $Ca(BH_4)_3^-$ and monocations $CaBH_4^+$, proposed to be the main electroactive species:

$$2Ca(BH_4)_2 \rightleftharpoons Ca_2(BH_4)_4 \rightleftharpoons CaBH_4^+ + Ca(BH_4)_3^- \qquad (1)$$

In the same work, molecular cluster calculations (embedded in a polarizable continuum model) indicated that the dimer is the second most stable conformation after the neutral monomer but lacked specific solvent interactions beyond the first coordination shell and any Debye screening from finite ion concentrations.

Interfacial characterization reveals the ready formation of solid-electrolyte interphases incorporating oxidized boron and even embedded calcium hydride[6]. And debate continues as to the presence or electrochemical relevance of the fully-solvated $Ca^{2+}$ dication[23,24,28,30–32,38].

In this work, we reveal intricate details of the population of species in the bulk of this electrolyte and striking differences within a nanometer of its electrode interfaces, which are further exaggerated by negative potential differences. We discuss the consequences of these predictions for functioning cells.

## Results

### In the bulk electrolyte

We employ free-energy analysis for a model of the bulk electrolyte at room temperature (RT), comprising two $Ca^{2+}$ ions (and four borohydride anions) dissolved in THF (Fig. 1a), using empirical force fields (see Methods and Supporting Information). The free energy surface (Fig. 1b) is sampled using metadynamics[39] with respect to two collective variables: the Ca–B coordination number, which controls the charge of complexes, and the Ca–Ca distance, which for a system with only two Ca ions, distinguishes between monomers and dimers.

This projection of the free energy landscape indicates a preference for dimer formation in competition with the entropy gains associated with free monomers. From the analysis of our molecular dynamics (MD) trajectories, Ca–Ca dimers are neutral complexes with the formula $Ca_2(BH_4)_4$ and come in two varieties: long dimers (63%) bridged by a single $BH_4^-$ and short dimers (37%) bridged by two anions (see below for more detail). Sampling local minima in the free energy surface allows us to parametrize a continuum model for chemical

equilibration at a finite concentration that includes a model of configurational entropy[40–42], as shown in Fig. 1c. In what follows, we will make explicit comparisons between the moderate (0.12 M) effective concentration of our MD simulations and a high concentration (1.65 M, close to saturation), which led to improved electrochemical performance in experiment[29,33,37]. The relative population of Ca–Ca dimers increases with concentration and is the most favored species above 2 M (although this is likely above the solubility limit). Conversely, the population of the neutral monomer complex, $Ca(BH_4)_2$, decreases with concentration. This might be expected from the perspective of solvent entropy—the need to coordinate more dissolved species at higher concentrations, which reduces entropy, can be somewhat offset by the formation of solute oligomers, such as contact ion complexes, dimers, etc., which require less solvent coordination.

It is notable that only a small percentage of the solvated population is predicted to be charged (<2% at low concentrations), dominated by the monocation, $CaBH_4^+$, with its complementary anion, $Ca(BH_4)_3^-$. Both bulk populations decrease with concentration. The free energy for the formation of fully solvated dications, $Ca^{2+}$, in this model, is too high (-19 kT) to support a significant bulk population at any concentration. These results provide some reordering with respect to static estimates, which predict the neutral monomer as most favorable, followed by a single (short) dimer conformation, the monocation, and the anion[29]. From the relative populations of each species, it is clear that neglecting the long dimer would lead to this conclusion. Furthermore, it is clear from RT sampling that there are multiple possible conformations for the dimer at finite temperatures (as we explore below). However, both sets of calculations agree that the fully solvated dication $Ca^{2+}$ is the least favored in this set of possibilities (1–1.2 eV higher in energy by static estimates[29], 0.48 eV from free-energy sampling at RT at an effective concentration of 0.12 M).

Our 2D free energy surface (Fig. 1b) reveals that direct interconversion of these solvated objects, by removal or addition of borohydride anions, is prevented by quite steep free energy barriers (24.4 kT or -0.6 eV). The easier path to disproportionation (forming charged species from neutral monomers) is through the formation of dimers, with the exchange of borohydride anions before dissociation into the complex monocation and anion as per Eq. 1 (with activation energies of 2.4–10 kT, Fig. 1b and Fig. S1).

At a minimum, this explains the prevalence of dimers in spectroscopic analysis of the bulk electrolyte (using EXAFS and Raman spectroscopy)[29] and possibly the observation of saturating ionic conductivity with increasing concentration as more neutral dimers form (prior to precipitation, Fig. 1c)[29,37].

The predicted concentration-dependence of neutral species is in great agreement with experimental reports[37]; and although we obtain a similar order of magnitude for the monocation, our results indicate that the monocation population actually decreases with concentration, opposite to experimental results. However, as we will see below, bulk concentrations of charged species may have little to do with the electrochemical performance which is dominated by interfacial phenomena.

Strikingly, we find that bulk populations of each complex are conformationally quite diverse. Our metadynamics simulations project the full free energy landscape onto only a few collective variables. However, multiple molecular conformations may satisfy these constraints (Ca–B coordination number and Ca–Ca distance, in this case). Data-mining techniques (dimensionality reduction, hierarchical clustering, and permutation-invariant alignment[20]—details in Methods and SI) applied to selective umbrella sampling of local minima in the free energy landscape reveal a rich variety of solvated isomeric structures, summarized in Fig. 1d.

For example, the neutral monomer, $Ca(BH_4)_2$, when additionally coordinated by three THF molecules (~60% of its population) adopts mostly bent borohydride arrangements with a large dipole moment

(~8 Debye, see Fig. 1d, isomer 3 THF-1 and SI Section 1). In addition, we found a significant portion (~30%) of monomers coordinated by 4 THF molecules with an axial borohydride arrangement and low dipole moment (~2 Debye, see Fig. 1d, isomer 4 THF-2)). These two structures had been proposed separately as the minimum energy structure from quantum-chemical cluster calculations[29] and molecular dynamics simulations[37], respectively. By contrast, the monocation occurs mostly with 5 coordinating THF molecules (see below), in agreement with previous reports[29].

We find that dimers are always neutral (with four borohydride anions) and are split in two main spatial configurations characterized as short (SD) and long (LD) dimers with average Ca–Ca equilibrium distances of 4.48 and 5.25 Å, respectively (Fig. S2). Furthermore, each was found to have sub-populations with 4–7 solvent molecules and, within those, several stereoisomers (Fig. 1d). Key differences between the long and short dimers are the presence of a single-anion or double-anion bridge and predominant 6 THF coordination or mixed 5–6 THF coordination, respectively. Short dimers are in excellent agreement with a previously proposed double-bridged dimer structure with a 4.4 Å Ca–Ca distance (from fitting to EXAFS data)[29]. The set of structures shown here expands upon and underscores the configurational flexibility of calcium[29]. And, based on our understanding of the efficient disproportionation pathways to form charged species via dimerization (discussed above), it makes sense that the dominant dimer conformations form from combinations of bent and axial borohydride arrangements of the neutral monomers (e.g., long dimer isomers 1, 3 and 7 with 6 THF molecules in Fig. 1d are bent-bent, axial-bent and axial-bent combinations, respectively).

Although all dimers here are neutral, each Ca ion within a given dimer may be locally coordinated by 1–4 anions, with 1 (long) or 2 (short) shared between them. Most commonly, we observe [3,2] or [3,3] anion arrangements for long or short dimers, respectively. Small populations of [1,4] dimers are found and are key to some interfacial

disproportionation processes discussed below and in Section 'Generation of active species'.

### At the electrode–electrolyte interface

What happens to species in the electrolyte as they approach an interface, such as the electrode surface? Firstly, as expected from simple statistical mechanics modeling of molecules at hard interfaces[43], the solvent, THF, adopts a layered molecular structure[36,44] near the surface with a dense layer (DL) at 3–6 Å, followed by a low-density 'gap', and an intermediate density layer (IDL) at ~7–10 Å from the surface, with 2.5, 0.3, and 1.5 times the bulk THF density, respectively (Fig. 2a). A third collective coordinate, calcium distance from the interface, sampled with metadynamics allows us to obtain the interfacial, 3D free-energy landscape (Fig. 2b), of which the most distant slice is the 2D bulk free-energy landscape in Fig. 1c. Dissolved species in the electrolyte near the electrode surface respect this underlying solvent layering with their free-energy minima distributed between the bulk, IDL, and DL, and separated by barriers, as indicated by minimum energy pathways of neutral and charged species approaching the interface in the 3D landscape (Fig. 2c, d).

A key observation is that the IDL defines an attractive basin for most species, especially dimers, and monocations, with minima in free energy that are lower than in bulk, implying that this is a narrow interfacial region for the enhanced concentration of solutes. Conversely, the DL defines a region from which solutes may be excluded due to additional free energy costs without the assistance of some applied bias (see below). This has some striking consequences for electrochemistry that will be apparent when discussing reductive processes (see below).

Expanding our continuum model to include the presence of an electrode and the sampled interfacial free-energy hypersurface[40–42] allows us to calculate the potential of zero charge (PZC), which is −1.8 meV at 0.12 M, with negligible variations in interfacial

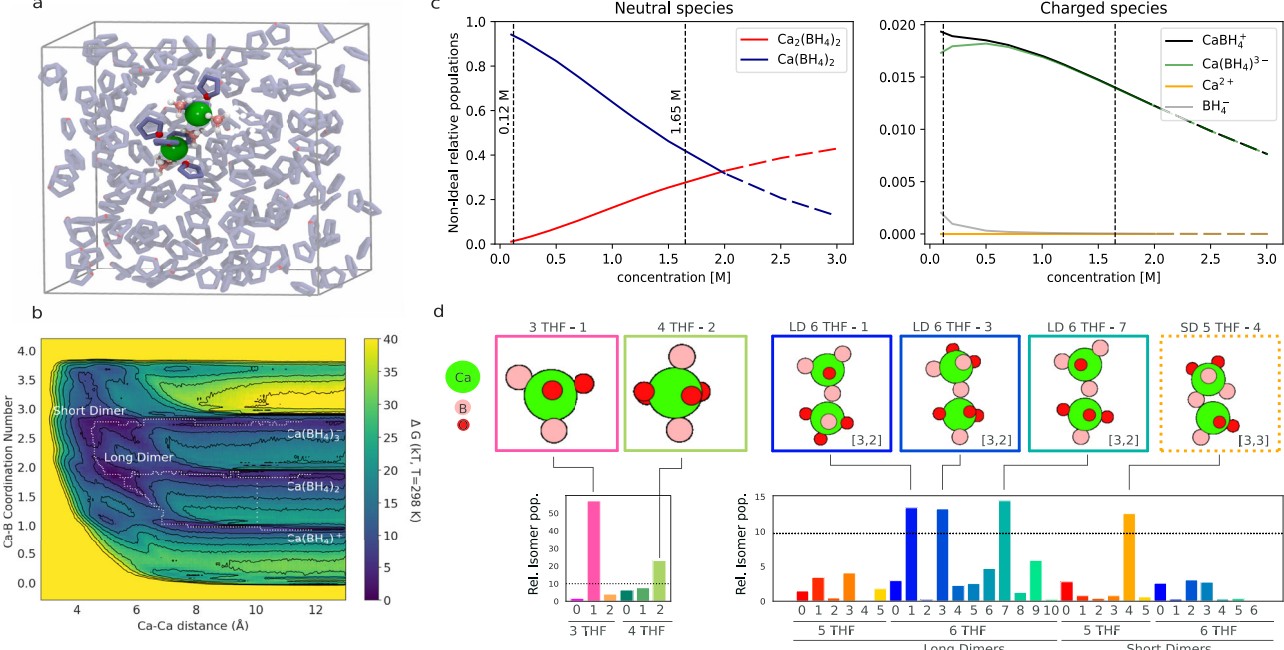

**Fig. 1 | Analysis of the bulk electrolyte. a** One time step sampled from our molecular dynamics model of the bulk electrolyte showing two $Ca^{2+}$ dications (green) and 4 $BH_4^-$ anions (B atoms pink) in THF (O atoms red) at room temperature (RT). **b** Free-energy surface sampled using metadynamics with respect to Ca–Ca distance and Ca–B(H₄) coordination number. Minimum energy pathways for dimer disproportionation from the neutral species, $Ca(BH_4)_2$, are indicated by dashed lines and were used to parametrize a continuum model to obtain **c** equilibrium bulk

populations of neutral and charged electrolyte species as a function of concentration. **d** Population analysis with respect to anion and THF coordination and conformation about Ca ions obtained via unsupervised learning, indicating the diversity of species umbrella-sampled from points on the free-energy surface corresponding to the neutral monomer (CN(Ca–B) = 1.9 and d(Ca–Ca) = 11 Å) and the long (L) and short (S) dimers (d(Ca–Ca) < 7 Å). Atomic structures of dominant structures (populations larger than 7.5%) are shown.

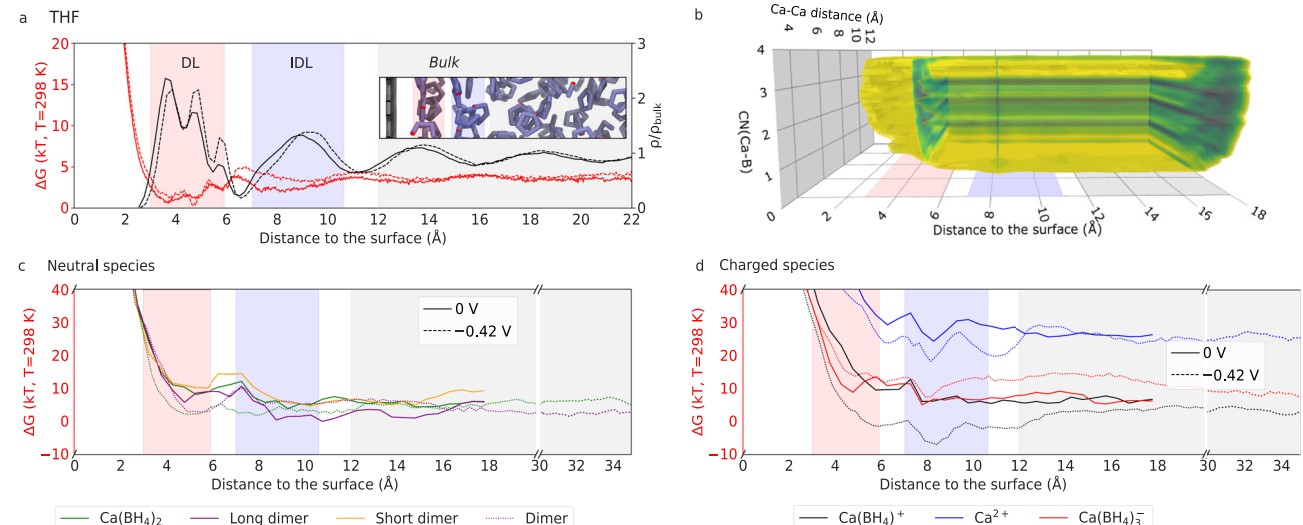

**Fig. 2 | Effect of solvent layering at the interface. a** The free-energy profile (red) of a single THF molecule in THF at RT with respect to distance from a graphite interface and the corresponding oxygen density profile (black), with slight adjustment (dashed lines) at negative bias. The dense layer (DL), intermediate density layer (IDL), and bulk regions are color-coded hereafter in red, blue, and gray. The inset shows a snapshot of the interfacial region sampled from molecular dynamics, indicating THF molecules lying flat against the graphite surface and somewhat constrained at the DL–IDL interface. **b** The 3D free energy landscape derived from metadynamics with respect to Ca–Ca distance, Ca–B coordination number, and distance from the graphite surface for the neutral interface (at 0.12 M). Minimum free-energy paths for (**c**) neutral and (**d**) charged species in the electrolyte arriving at the interface from the bulk under zero (solid) and negative (dashed) bias conditions. Sudden jumps in the free-energy profiles reflect variations with respect to other collective variables in the free-energy landscape. Under negative bias, the long and short dimers are not distinguished in our sampling. In general, negative bias stabilizes charged species at the interface.

concentration between open circuit potential (OCP) and PZC conditions. The metadynamics simulations used in this section, with zero excess charge, are thus representative of an open-circuit system.

Using our re-parametrized continuum model, we find that, next to this unbiased non-interacting electrode (Fig. 3a), concentration effects are even stronger than in bulk. At 1.65 M, the IDL is dominated (~86%) by dimers (long dimers in particular, 55%), with reduced populations of neutral monomers (~11%) and a very slight increase to 2% in the monocation population (Fig. 3b). Around 30% of the interfacial population is in the DL, where monocation population rapidly declines and dimers dominate (~92%)—especially short dimers, in contrast to bulk and IDL populations. At low concentration, 0.12 M, only 13% of the population is in the DL; neutral monomers dominate the DL and IDL; dimer population is limited to 15 and 18%, respectively; and there is a slight accumulation of monocation at the IDL (7%).

Unsupervised clustering analysis of structures from umbrella-sampling trajectories at the IDL and DL ($z = 8.25$ and $5.75$ Å, respectively) reveals a reduced number of favored dimer isomers compared to the bulk, with one given isomer making > 20 % of the population in each case (Fig. 3c, d). In the IDL, this is a bent-axial long dimer with 6 solvating THF molecules (indexed as Isomer 7 or LD 6 THF-7), already a favored species in the bulk. In the DL, on the other hand, a short dimer also with 6 THF molecules (SD 6 THF-2) dominates. Furthermore, we find that the orientation of dimers at the interface is discretized (Fig. 3e, f). Favored IDL dimers have mostly flat orientations of the Ca–Ca vector relative to the graphite surface, with two dense layer THF molecules involved in solvation. Similarly, the dominant dimer orientation in the DL is flat (i.e., in a plane parallel to the surface), with both calcium ions embedded in that dense region and with two THF molecules from the IDL contributing to solvation. Additionally, perpendicular dimers form at least 11% of the DL population. Isomer 4 of the short dimer with 5 THF molecules (SD THF 5–4) is an asymmetric dimer, with calcium ions solvated by 2 and 3 THF molecules that sit in the DL and on the edge of the IDL, respectively (Fig. S3). Note that the specific conformation of these coordination complexes defines their effective dipole moment, which may, in some cases, be orthogonal to

the Ca–Ca vector of the dimer. An example in the DL is the favored isomer SD 6 THF-2, with its Ca–Ca vector parallel to the surface but three $BH_4^-$ sitting closer to the interface, near the corresponding DL free energy minimum for isolated borohydride in THF next to graphite ($z = -4$ Å, see Fig. S4). Dipole orientation will be discussed in more detail below in the context of biased interfaces.

The closest approach of fully solvated ions (as either dimers or neutral monomers at OCP) takes place in the IDL, which, by some definitions, is analogous to the Outer Helmholtz plane (OHP). Similarly, the DL is a close analog of the inner Helmholtz plane (IHP). One key distinction from textbook definitions of the IHP (usually for aqueous electrolytes) is that THF layering in the dense layer leads to a flat orientation of THF molecules with respect to the electrode surface, and we did not observe any large reorientation of its dipole with changes in surface charge (below). In this electrolyte, only the dipole moments of dissolved complexes exhibit reorientation in the DL. Finally, a lack of solvent layering more than 12 Å from the electrode maps onto the diffuse layer.

**Negatively biased interfaces**

So far, it seems that $Ca(BH_4)_2$ in THF is a poor electrolyte, with only a small fraction of the salt concentration (2% for both 0.12 M and 1.65 M) present as charged species in the bulk, albeit with a noticeable enhancement to 7% in the IDL for a concentration of 0.12 M at OCP. Otherwise, this electrolyte is dominated by neutral species (monomers and dimers). This is consistent with the previous discussion of undissociated neutral species as dominant in Ca-based electrolytes with boron-containing anions[11]. However, the bulk free energy landscape (Fig. 1c) indicates that interconversion of species is possible (more details below), and some equilibrium exists between charged and neutral species. We explore the impact of biased/charged electrodes on the population of solutes by evenly distributing opposing charges on either face of the two-layer graphite electrode model, which, under periodic boundary conditions, polarizes the electrolyte. We considered two specific charge states with (1) 0.065 and (2) 0.13 e/nm$^2$ (labeled CS1 and CS2, respectively). These surface charge densities

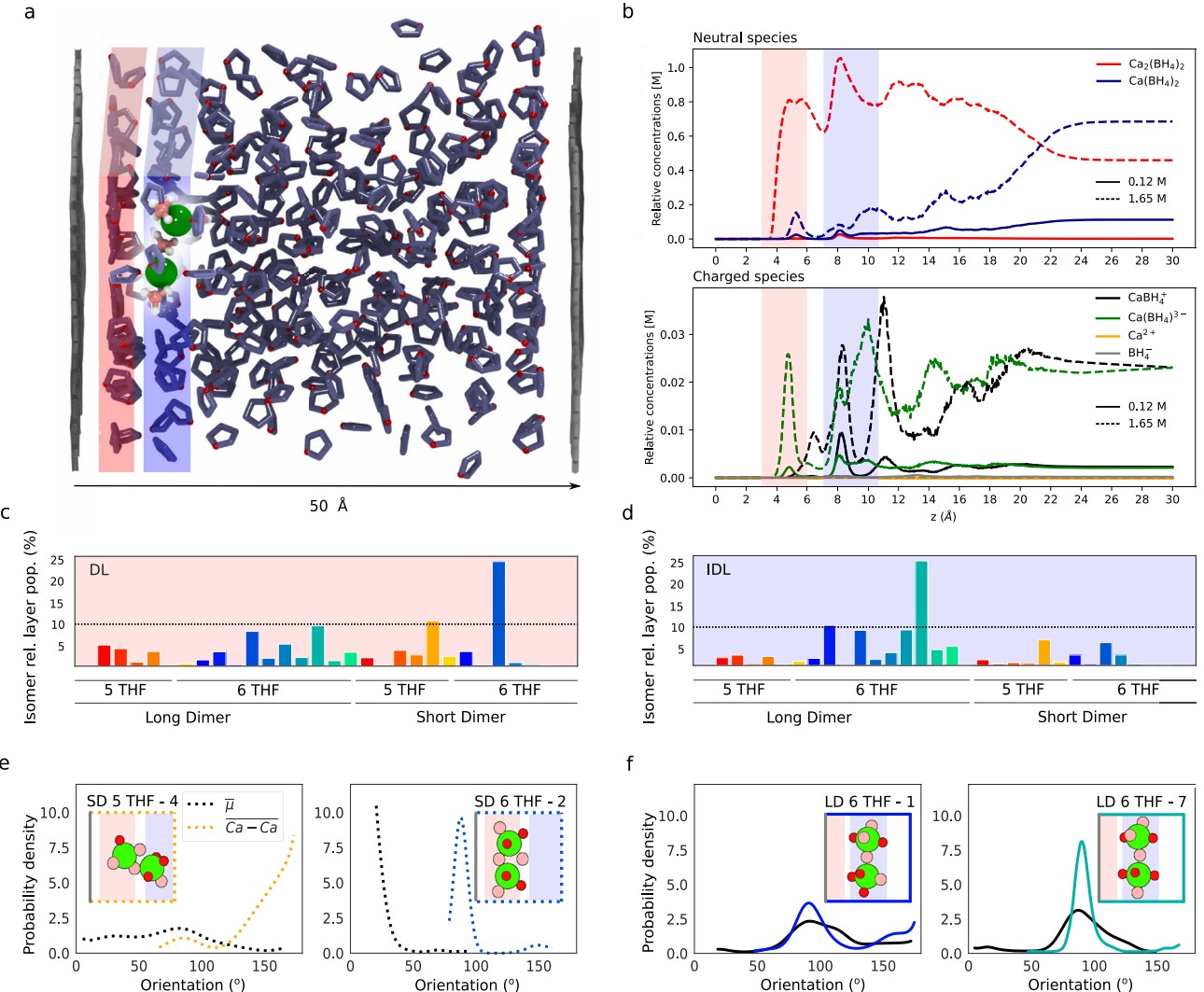

**Fig. 3 | Distribution of species at the neutral interface.** A molecular model of the simulation box, obtained from a snapshot of the equilibrated MD trajectory (**a**), shows a dimer at the IDL (in blue) near the DL (in red). The relative concentration of neutral and charged species as a function of distance from the electrode (**b**) was calculated with the continuum model for total bulk concentrations 0.12 and 1.65 M. The DL and IDL regions are marked with red and blue rectangles. Unsupervised clustering analysis of umbrella sampling trajectories at the DL and IDL ($z = 5.75$ and 8.25 Å) show the relative populations of dimer isomers per layer (**c**, **d**).

Representative structures have discrete orientations, which depend on the layer (insets in **e**, **f**). Color lines show the orientation of the Ca–Ca axis with respect to the surface normal, where 90° is parallel to the surface, as shown in IDL dominating dimer LD 6 THF-7 (also in **a**), while DL dimer SD 5 THF-4 is mostly perpendicular. Dipoles also have discrete orientations with respect to the surface normal for given isomers (shown in black, dotted lines) and are roughly aligned to the Ca–Ca axis in long, flat dimers (IDL) or perpendicular, as in DL dimer SD 6 THF-2.

correspond to potential differences of −0.26 V (CS1) and −0.42 V (CS2), calculated using our continuum model (see Methods) assuming a simulated bulk concentration of 0.12 M. (Note that at higher bulk concentrations, with a shorter Debye screening length, this estimated potential difference should be even smaller—at 1.65 M: −0.20 V (CS1) and −0.35 V (CS2).) These potentials are sufficient to draw monocations into the dense layer.

From Fig. 2a, we see that the solvent layering remains practically unchanged upon charging the electrode. At high concentration (1.65 M), a third of the interfacial population is in the DL. Dimers are still the most favored species in the DL and IDL in both charge states (Fig. 2a, b). With the increase in the bias potential, some charged species become strongly stabilized at the interface. Specifically, the monocation, $CaBH_4^+$, partly displaces the neutral dimers and monomers (Fig. 4b) to become up to 27 and 21% of each layer's population, respectively (compared to 0.3 and 2% in the unbiased electrode). The concentration dependence already seen at the unbiased electrode becomes far more striking at negative biases. Low concentration

(0.12 M) leads to a sparser DL, which constitutes only 16% of the total interfacial population. Both DL and IDL are completely dominated by the monocation (98 and 92%, respectively, for CS2), while the dimer population declines less than 1% in both. As in the unbiased electrode, the bulk population at low concentration consists mostly of neutral monomers (95%).

Notably, the peak concentration of monocation in the DL sits near its external edge, at ~6 Å. At this potential, the approach of the monocation to the electrode through the DL would have to be an endergonic process, requiring 5.7 kT of free energy overcoming a barrier of 7.5 kT. However, this is some improvement over the case of the neutral electrode, which presents a barrier of 9.9 kT to enter the DL, with a required input of 9.1 kT of free energy.

We can understand these free energy costs by following the evolution in solvation and associated dipole orientation of the monocation. Unsupervised learning analysis of various umbrella sampling trajectories in the IDL, at the edge of the DL, and in the DL ($z = 8.25$, 5.8, and 4.8 Å, Fig. 4c–e) reveals that fivefold THF

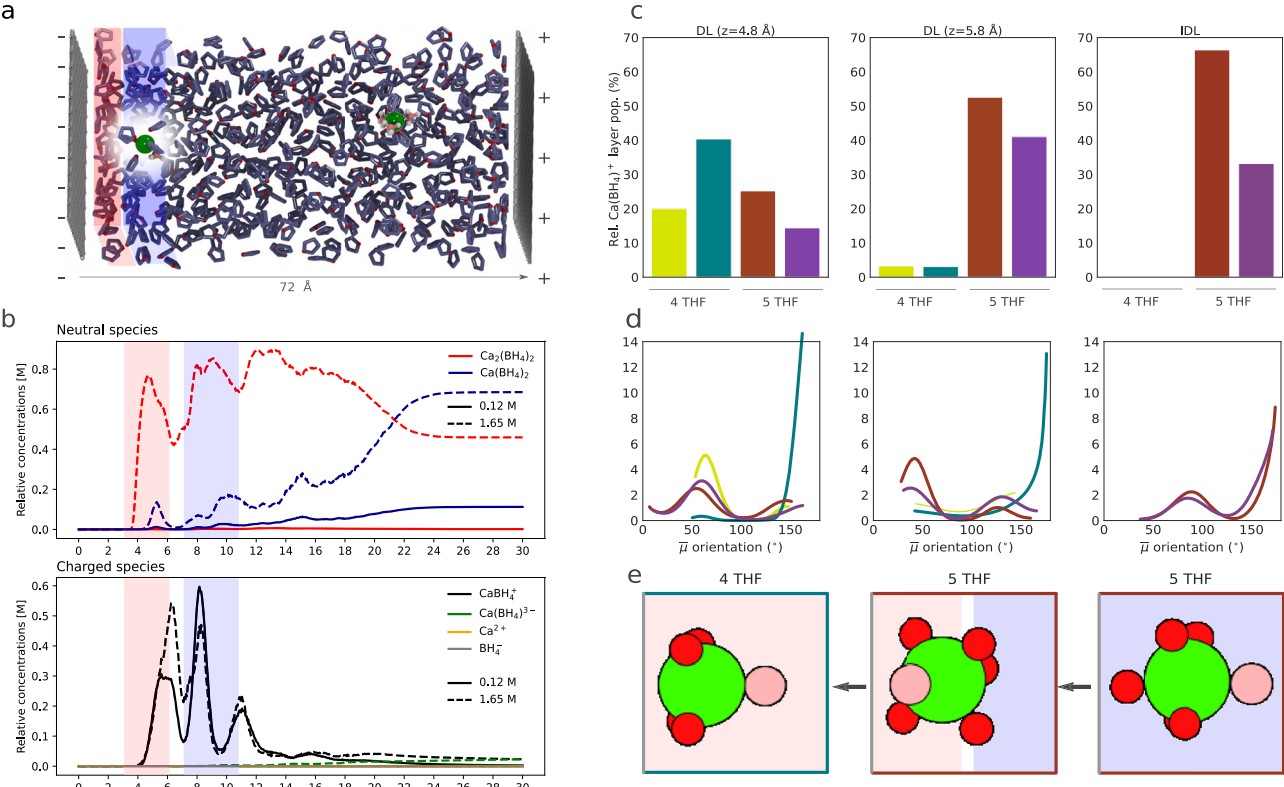

**Fig. 4 | Populations at a biased interface.** Snapshot obtained from the equilibrated trajectory of charged state 2 ($\varphi(0) = -0.35$ V) with highlighted layering (red/blue for DL/IDL) at the negative interface showing the IDL solvated monocation (**a**). The relative concentration of neutral and charged species as a function of distance from the negatively charged electrode (**b**) at potential differences corresponding to CS 1 and CS 2 for a bulk concentration of 1.65 M was calculated using the continuum model. Again, the DL and IDL regions are marked with red and blue rectangles. Unsupervised clustering analysis of Umbrella Sampling trajectories of the monocation at the IDL, close to the edge of the DL ($z = 5.8$ Å) and close to the center of the DL ($z = 4.8$ Å) (**c**) classifies the structures into two isomers of the fivefold and fourfold THF coordinated monocation, which differ by slight changes in the local geometry of the coordinating THF molecules. The corresponding Ca–B dipole orientation with respect to the surface normal ($\vec{\mu}$), with 180° corresponding to B pointing away from the surface, indicates discrete orientation at the interface (**d**). Representative structures of the main isomers at each layer in their most likely orientation, with the surface on the left side (**e**).

coordination dominates the IDL population, with the $BH_4^-$ anion pointing away from the surface—as one might expect given the direction of the electric field at the negative electrode. However, this dipole reorients at the edge of the DL, likely due to mixed DL/IDL solvent coordination, and a small fourfold THF coordinated population appears. Ultimately, in the center of the DL, fourfold coordination dominates, with the dipole of the predominant isomer pointing away from the surface again.

This complicated and costly path for the monocation to reach the electrode further emphasizes the important role of the THF solvent layering and specific coordination in determining the electrochemical activity. By the same token, the fully-solvated dication, $Ca^{2+}$, with its somewhat rigid first solvation shell, is still too unfavorable to define a noticeable population at these bias potentials despite its higher electrostatic charge. Stabilization of $Ca^{2+}$ species to reach non-negligible concentrations is only achieved at relatively large (yet still non-reductive) potential differences in the inner DL (<−1.1 V at 1.65 M), where preferred isomer structures are undercoordinated and flattened (see below).

### Generation of active species

Thus far, it seems that the monocation, $CaBH_4^+$, is the strongest candidate for the electroactive species in this electrolyte, given its high population in the vicinity of the electrode upon negative charging. The fact remains that this poor electrolyte (only 2% of species in solution are charged, as mentioned above) must supply the DL and IDL with monocations in the first place via some disproportionation

mechanism from neutral species (most likely dimers) and replenish the same species while electroreduction and electrode deposition consumes them. Any barriers in this supply chain should be evident in the kinetics of the electrochemistry as an observed deposition overpotential or associated rate limitations in charging cells with this electrolyte. The much smaller concentration of dimers in low-concentration electrolytes may be partly behind the loss of performance observed in experiments at concentrations below 0.5 M—which exhibit lower current densities and lower Coulombic efficiencies[29,33].

To shed light on the underlying processes, we approach the generation of charged species (monocations) as a two-step process involving dimer reorganization and disproportionation. As we show below, the most stable or prevalent dimer species are not readily disposed to disproportionate. Some molecular rearrangement of borohydride anions and solvent molecules is first required. The subsequent disproportionation follows two major pathways, which can occur in the bulk electrolyte or at the interface and which we have investigated both at neutral and negative biases.

In the neutral cell, increased dimer concentrations in the IDL are readily explained through the analysis of the free energy landscape (Fig. 2). The minimum energy path (MEP) for disproportionation in the presence of the interface indicates that neutral species in bulk can easily flow, without a significant free energy barrier, into the IDL. Migration from the bulk to the IDL is essentially barrierless for all species except $Ca^{2+}$. As summarized in Fig. 5a, to prepare for disproportionation of the most favorable dimers, some reorganization

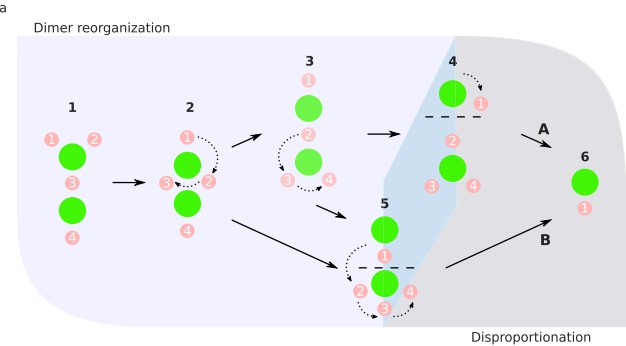

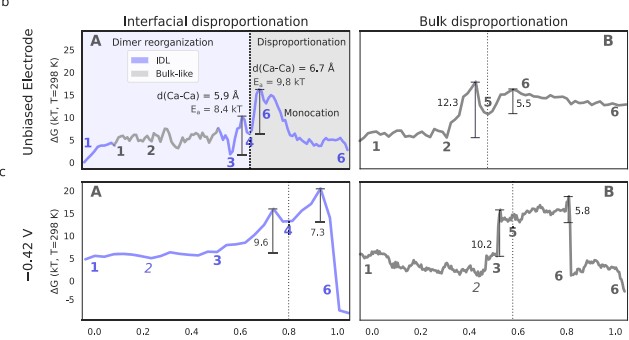

**Fig. 5 | Disproportionation pathways.** Scheme depicting the main two steps of dimer disproportionation **a**: reorganization (**1**–**5**), followed by separation into ions (**6**) through distinctive paths A and B. Minimum energy pathways for disproportionation in the IDL and bulk for **b** neutral and **c** negatively charged (CS2) electrodes, obtained from the 3D free-energy landscapes. Color-coding indicates distinctions between processes in the IDL (blue) and the bulk electrolyte (gray).

into an intermediate (less stable) dimer is required, and ultimately Ca–Ca separation into ionic species follows two main paths, A or B.

For Path A, the dominant long dimer configuration (**1** in Fig. 5a) can undergo a low-barrier reorganization through a short dimer (**2**), to another long dimer with similar coordination of Ca ions with borohydrides (**3**). Then, the Ca–Ca distance increases (**4**) up to 6.7 Å at the transition state so that the nascent monocation Ca ion can increase its number of coordinating THF molecules from the original 3–4 to the preferred 5 upon full dissociation. Path B branches from the short (**2**) or intermediate long (**3**) conformations, through further borohydride and solvent reorganization, to a higher energy dimer (**5**), with a [1,4] anion coordination, which then disproportionates to produce the monocation (**6**).

Figure 5b, c outlines the MEP for disproportionation at the electrode interface or in the bulk electrolyte under neutral or negative bias. Overall, at the electrode interface, in the IDL, disproportionation follows Path A, whereas Path B is preferred in bulk, likely due to configuration **5** being more favorable in bulk than at the interface (Fig. S5).

We find that interfacial (IDL) disproportionation at zero bias via Path A is favored since it has slightly lower barriers ($E_{a,3\rightarrow4} = 8.4$ kT and $E_{a,4\rightarrow6} = 9.8$ kT) and a lower free energy cost than bulk disproportionation via Path B (Fig. 5b). This is in agreement with the slight increase in monocation population observed at the IDL in Fig. 3 (Slight variations in barriers between bulk and interface models can be due to differences in the collective variables and grid-spacing employed in our metadynamics simulations, SI Table S1).

At the negatively charged electrode, we have seen (Fig. 4) that the monocation is favored in the DL and IDL, displacing the previously dominant dimers (neutral monomers) at high (low) concentration with increasing negative charge on the electrode (CS2). Due to the size limits of our metadynamics simulations, we may well expect that the bulk thermodynamics are somewhat different from those under neutral conditions. However, disproportionation still follows Path B in our simulations (Fig. 5c), albeit with an additional step involving the formation of a long dimer conformation (**3**). Similarly, Path A is still preferred in the IDL. Although disproportionation occurs in bulk, barrierless pathways to the IDL suggest that dimers can approach the charged IDL and undergo disproportionation quite favorably. Therefore, upon negative charging, the concentration of monocations should increase in the IDL, based on favorable free energy and necessary – highly concentration-dependent– supply from the local (IDL) dimer population via interfacial disproportionation.

## Reductive processes and the seeds of the SEI

The stabilization of the monocation in the DL at negative bias results in a significant increase within the overall interfacial (DL plus IDL) population of this species—from ~1% (6%) in the unbiased electrode to ~23% (94%) for CS2 at 1.65 M (0.12 M). Structural analysis indicates that these monocations have their anionic end pointing away from the negative surface and have lower solvent coordination.

Using our interfacial continuum model, we explore increasingly negative electrode potentials at or slightly above the thermodynamic Ca deposition potential (−2.25 V, see Supplementary Information 3). We limit our discussion here to an electrolyte concentration of 1.65 M. Strikingly, a highly localized dication population appears in the inner DL (3–5 Å, Fig. 6a). Unsupervised learning analysis, this time including whole solvent molecules in the coordination sphere, reveals that dication coordination at these short distances consists mostly of flattened, under-solvated (5 THF) structures. The more dominant monocation population is pushed slightly outwards by that first dication layer. This begs the question as to which species contributes the most to the electron transfer rate.

Electrochemical activity is a strong (exponentially decaying) function of the distance of the reducible species from the electrode. It is also very likely enhanced by reduced cation coordination, i.e., favoring reduction of species that are less thermodynamically stable[17]. However, this increased thermodynamic cost would also lower the local concentration of the same species. Additionally, the magnitude of the required reduction potential of a given species can increase with proximity to the interface due to the stabilization of positively charged species by the negative electrode potential. This trade-off between species availability (concentration), distance to the electrode, and reductive stability is key in determining each species' contribution to the electron transfer rate.

The reduction potential of each isomer was calculated using a combination of quantum-chemical calculations, free-energy sampling, and potential-dependent concentration profiles from the continuum model. Contributions to an effective electron transfer (ET) rate were estimated based on distance-dependent tunneling decay, reduction potential, and isomer concentration (see Methods). Note that the concentration (more precisely, activity) dependence of the ET rate effectively follows the Nernst equation, although we are not modeling an equilibrium process here. Based on our calculations, we see 100- to 10,000-fold increases in the ET rate as we increase the electrolyte concentration from 0.12 to 1.65 M (see Fig. S6). In what follows, we focus on results for 1.65 M electrolytes.

We further simulated the effect of SEI growth on ET rate within the continuum model by including a progressively wider dielectric spacer under the assumption that electrons could tunnel through it with the same decay constant. In Fig. 6a, we can see that the effect of holding incoming positively charged species farther away from the negatively charged electrode is to reduce their effective concentrations in the interfacial (dielectric-electrolyte) region due to the decreasing potential difference in this region with increased spacer width. Most

noticeably, this removes the interfacial population of dications that we observed at the pristine electrode.

Increases in the magnitude of the negative electrode potential above the thermodynamic deposition potential (labeled as over-potential here) have only minor effects on these concentration pro-files. However, the species-dependent reduction potentials and the combined estimate of the ET rate are strong functions of this over-potential. In fact, the presence of a dielectric spacer that is close in width to the ET decay length (approximated as 1 nm here) appears to lower the required overpotential to reach the same effective ET rate, as shown in Fig. 6b. For example, the pristine electrode exhibits an effective ET rate of ~$10^2$ at a 0.25 V overpotential, while the same or higher rate is possible with dielectric spacers of 0.5–1.5 times the ET decay length at a smaller 0.10 V overpotential. Note that the over-potentials discussed here are thermodynamic in origin.

With no dielectric spacer, the main contributing species to the ET rate are monocations coordinated by 4 THF molecules in two specific arrangements (4 THF 0 and 4 THF 2), nominally undercoordinated and flattened with respect to their bulk electrolyte coordination, located at the outer edge of the DL (beyond 5 Å). A representative molecular structure is provided in the inset of Fig. 6c (see the rest in Fig. S7). Interestingly, these monocations are not as close to the electrode as they could be because of the presence of dications stabilized by the strong negative potential difference close to the electrode. However, the same dications require a larger overpotential to be reduced due to

the stabilization provided by this potential difference and thus do not contribute significantly to the effective ET rate.

With a 5 Å spacer, the potential difference at the electrolyte interface is now lower, and the less thermodynamically favorable dication population disappears. The electroactive species in the DL region are now exclusively monocations, which can occupy the inner DL and be reduced at lower overpotentials due to a decrease in ther-modynamic stabilization by the smaller potential difference this far from the electrode.

The final outcome of this analysis is that the undercoordinated monocations in the DL are the dominant contributors to the electro-chemical activity, with negligible contributions from the IDL, as shown in Fig. 6c. Therefore, we would conclude that Ca electrodeposition is defined by inner sphere processes in this electrolyte. And surprisingly, the presence of a thin SEI layer may reduce the required (thermo-dynamic) overpotential for electrodeposition, indicating the impor-tance of activating such electrodes during the first charge.

## Discussion

Based on our analysis of the Ca(BH$_4$)$_2$|THF electrolyte and its interfacial speciation, we can propose the following phenomenology that may explain existing observations and provide guidance and interpretation of future characterization efforts.

First and foremost, competition between the solvent, THF, and the anions, BH$_4^-$, for coordination of the dication, Ca$^{2+}$, leads to an

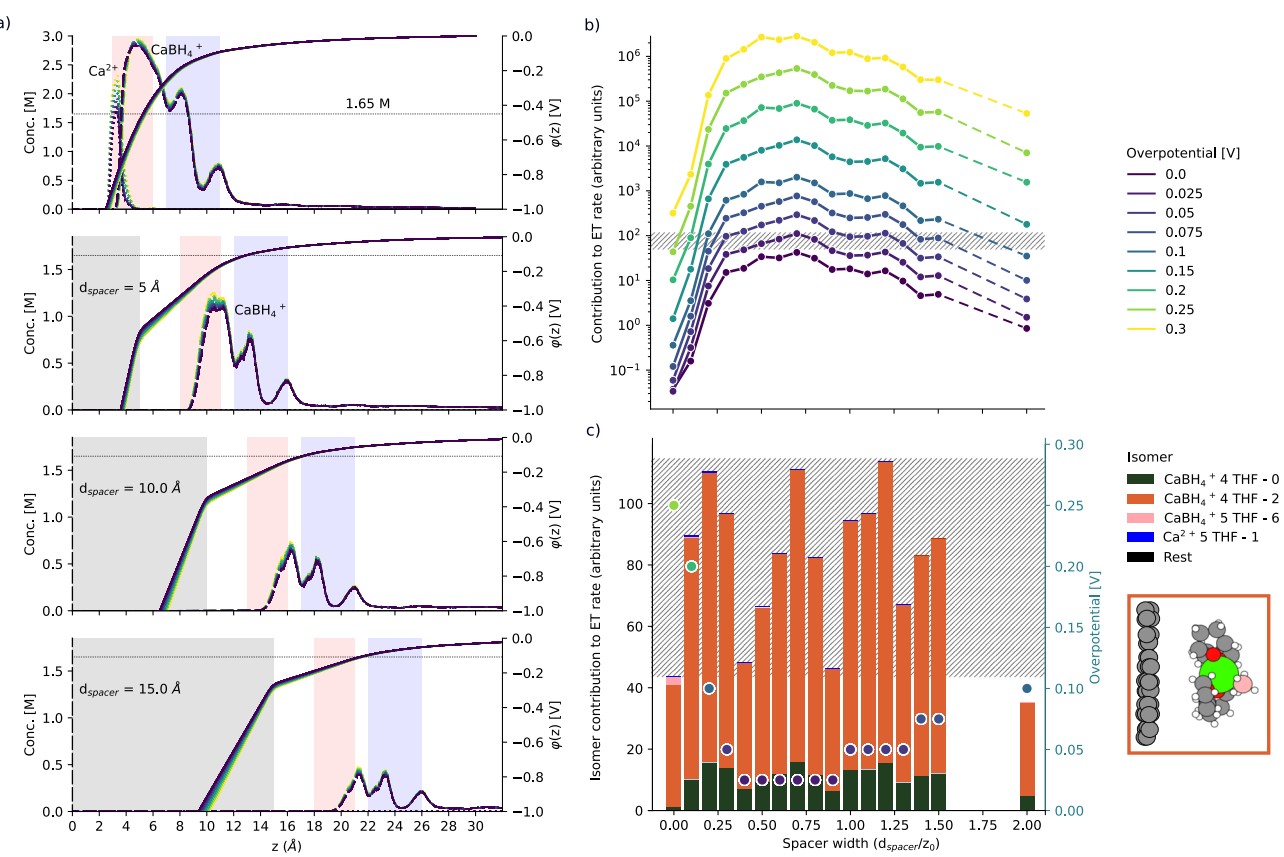

**Fig. 6 | Reduction at the negative electrode. a** Interfacial concentration profiles of monocation and dication as a function of overpotential at 1.65 M for the bare electrode (top) and with dielectric spacers of width $d_{spacer}$ = 5, 10, and 15 Å. **b** Contributions to the electron transfer rate integrated over the interfacial region for specific overpotentials as a function of spacer width (expressed as a fraction of the tunneling decay length, $z_0$ = 1 nm). The experimentally observed overpotential on the first charge is 0.25 V[23]—which we assume corresponds to our bare electrode model (without a dielectric spacer). A band of similar ET rate magnitudes to that at

0.25 V is indicated by a horizontal hatched pattern, highlighting that lower over-potentials combined with finite dielectric spacer widths can lead to similar ET rates. **c** A closer view of this effective ET rate regime indicates the relative contributions of individual dication and monocation isomers in the DL at the corresponding over-potentials (colored dots). IDL contributions are negligible and thus not plotted. The inset on the right shows the structure of the dominant contributing isomer: a flattened, undercoordinated monocation complex, 4 - THF 2, as obtained via unsupervised learning analysis, including the entire solvation sphere.

electrolyte dominated by neutral species, mostly monomers, $Ca(BH_4)_2$, with an increasing prevalence of neutral dimers, $Ca_2(BH_4)_4$, with increasing concentration. The formation of charged species is facilitated by the disproportionation of dimers rather than the more thermodynamically expensive direct dissociation of neutral monomers.

Strong solvent layering defines two interfacial regions within the first nanometer of the electrolyte: a dense layer inside 6 Å and an intermediate density layer from 7 to 10 Å. These layers present mildly attractive or repulsive thermodynamic potentials that modulate the interfacial population of all species relative to the bulk electrolyte, even in the absence of electrode biasing. With increasing concentration, we see enhancements in the populations of various species in these interfacial zones, which has important consequences both for the availability of these species for electrochemical processes (the monocation $CaBH_4^+$) and their replenishment (via dimer disproportionation).

Overall, this is a poor electrolyte, with very low concentrations of charged species in bulk solution. However, it is "activated" at biased electrodes within this narrow interfacial region, wherein large relative populations of charged species emerge, especially with increasing concentration. Therefore, meaningful characterization of the electrochemical activity of this electrolyte requires operando measurements that are sensitive to within a nanometer of the interface. The rich isomer subpopulations with distinct orientations in the IDL make it an excellent playground for interfacially sensitive, polarization-dependent spectroscopies that can capture these differences. Furthermore, the bias-dependent switch in the local population from oriented dimers to monocations should be observable with chemically sensitive vibrational[45–47] and electronic probes[40,48,49].

With increasing negative potentials up to and exceeding the thermodynamic potential for Ca deposition, we find that the pristine electrode may require a significant overpotential to register a deposition current. However, the presence of a thin SEI (introduced here as a dielectric spacer) can reduce this overpotential significantly due to decreased electrostatic stabilization of charged species. This may explain the observation in previous work that electrochemical activity, viz. plating of Ca using this electrolyte, carries a first, short duration overpotential of -250 mV, followed by a -100 mV overpotential in subsequent cycles[23].

In addition, we find strong dependence in the electron transfer rate on the overall bulk concentration, primarily due to a decrease in the overpotential required for each species as its relative concentration increases (as seen in the Nernst equation).

Strong solvent and anion coordination of electrochemically active species, which has dominated our analysis, is very likely the source of solid-electrolyte interphase (SEI) formation due to electroreduction and decomposition of these ligands at the interface, as highlighted recently through electron microscopy[6]. Similarly, for other multivalent ion electrolytes, we know TFSI is both inherently and electrochemically unstable for Mg[50] and for Ca[51].

Increased electroreductive stability in large anions may be afforded by considering non-cation-coordinating species. For example, closoboranes have been studied with Mg in tetraglyme[52]. These bulkier anions may also lead to better electrolytes overall (for example, preventing the formation of dimers), readily producing charged electroactive species. Although, strong solvent coordination, as we have seen for the fully-solvated $Ca^{2+}$, may still lead to significant overpotentials for electrodeposition and associated solvent decomposition and SEI growth.

From the computational perspective, we have highlighted the value of free-energy exploration through combined molecular dynamics, continuum modeling, and unsupervised learning to reveal the complexity of this nominally simple electrolyte and tried to connect our simulations to observed electrochemical behavior and characterization. However, as in all theoretical models, our study has some inherent limitations. The complexity of the system and the time scales required to explore different coordination complexes required the use of empirical force fields rather than ab initio methods. The finite number of dissolved species in our MD simulations mimics only low concentrations, which we extend to higher concentrations only at the level of our continuum model. Accurate inclusion of Debye screening from finite ionic concentrations and dielectric screening due to solvent or anion polarizability[53] (which we approximate only by charge rescaling) should be considered in future studies.

The notable outcome of this study is a reinforcement of the notion that performant nonaqueous multivalent electrolytes (with high ionic conductivity and low overpotential) have competing requirements for multivalent ion coordination: to be strongly coordinated to keep the salt dissolved and the ionic conductivity high while not so strongly coordinating that the ion cannot break free from solvation during electrodeposition. The need for strong solvent coordination is driven by competition with favorable ionic bonds between counterions. If we want to maintain the advantages of earth-abundant, multivalent ions, then one option would be to switch to larger anions without specific coordinating moieties. However, this study highlights that there may be other options to consider that sideline the isolated multivalent ion altogether, which may even be irrelevant in terms of electrochemical activity. Firstly, that coordination with counterions may actually help bring active species closer to the electrode and that the strength of solvent coordination can dictate which species approaches the electrode closest. Secondly, incomplete dissolution and the involvement of oligomers (dimers in this case) could be key to improved electrochemical activity, albeit balanced by some power limitations due to reduced bulk ionic conductivity and the need to regenerate electroactive species through a disproportionation equilibrium. Clearly, we have more work to do, both in terms of understanding the specifics of the reduction of these clusters and their dissociation in the reduced state leading to Ca deposition; the potential negative side-effects of reductive instability of the coordinating species, already connected to SEI formation;[6] and the ultimate origins of measured overpotentials and currents in experiments. However, we have highlighted the importance of free energy sampling to attack such complex problems, even within relatively ideal conditions and contexts, and we look forward to seeing more studies of this kind in the future.

## Methods
Metadynamics sampling (MTD) with a classical force field was used to obtain free-energy landscapes as a function of $n$ collective variables. Equilibrium populations at critical points of a given landscape were then collected with Umbrella Sampling (US). Structural analysis of the US trajectories was then performed with a Python-based unsupervised learning protocol. In order to explore the effects of finite concentration and bias potential at the electrode, a continuum model was developed within the modified Poisson–Boltzmann equation framework with the inclusion of chemical equilibrium parameterized from the sampled free-energy differences. Finally, reduction potentials were calculated using the results of the continuum model and quantum-chemical calculations of the structures obtained from the unsupervised learning analysis.

### Free-energy sampling
Metadynamics free-energy sampling[39] was carried out using the COL-VARS module[54] implemented in LAMMPS[55]. Systems (see Table S1 for a full list) were generated using Packmol[56]. Concentration values were chosen to avoid forcing aggregation. Dimerization free energy surfaces show that at Ca–Ca distances larger than -10 Å, the free energy converges with respect to Ca–Ca separation. That is the cutoff we consider between "dissociated" and "aggregated", giving an effective

radius of 5 Å for the first coordination shell of the dication. Additionally, g(r) shows that the second solvation shell (O-THF) settles at around 6 Å. Assuming an optimal close-packing of ions, we deduce that solvent-separated ion pairs would be unavoidable at concentrations above 0.5 M for a 6 Å radius and 0.7 M for a 5 Å radius. Hence, for the purposes of molecular dynamics sampling, we selected concentrations ≤0.03 M, well below these limits. System equilibration consisted of conjugate-gradient minimization to avoid steric clashes, followed by a short NVT warm-up to room temperature (298 K) using a 1 fs timestep. Box-size equilibration was achieved by continuing the trajectory under NPT conditions at 1 atm with a 2 fs timestep. A final NVT step with the equilibrated lattice parameters (20 ns) was sufficient to bring the systems to equilibrium. Force-field parameters[57,58] were validated in our previous work on the same system[20]. The graphene was frozen in place by setting the forces to zero in order to ensure neutral and charged simulations were comparable. This MD setup was kept for the MTD and US simulations. Metadynamics calculation parameters—the width of the grid along a collective variable (W), height of the Gaussian "hills" used to bias the potential (H), and the frequency at which they are added (Freq)—can be found in Table S1 and were chosen to ensure convergence, namely, that the simulation reached the diffusive regime in the given collective variable space[36]. Faster completion times were achieved by taking advantage of multiple-walker metadynamics, which allows for parallelized sampling among several trajectories (replicas) that update their biased potential at a given frequency (RepFreq) with the total biased potential. Despite this, the grid used in the three-dimensional MTD calculations was necessarily coarser. The minima explored here tend to be separated by more than 0.7 Å (e.g., between the long and short dimer or between solvent layers), which is larger than the coarser grid resolution. In order to keep cell neutrality and consistency with the neutral simulation, charge states 1 (CS1) and 2 (CS2) were generated by adding equal and opposite charges ($\pm 2\,\mu$C/cm$^2$) distributed evenly among two graphene layers, emulating a positively charged and a negatively charged electrode.

Free-energy sampling was performed with combinations of the following collective variables: the distance between the calcium and the center of mass of the top graphene layer (dZ); the coordination number between the calcium and the boron atom in a given BH$_4$ (CN(Ca−B), $r_0 = 3.8$ Å); and, to track dimerization, the distance between two calcium atoms (dCa) or a Ca−Ca coordination number (CN(Ca−Ca), $r_0 = 6.5$ Å).

Note that the state free-energies in Fig. 1b were obtained by thermodynamic integration of our 2D free energy surface (Fig. 1c) over regions delimited by given collective variable values (e.g., for the Ca(BH$_4$)$_4^{2-}$ state, a Ca−Ca distance larger than 7 Å and a CN(Ca−B) larger than 3.5). Collective variables apply to only one of the calcium atoms, and hence, the free energy is averaged over all (remaining) possible coordination/distances for the other atom. Hence, the room-temperature Boltzmann probability speaks of the likelihood of finding a state formed by the constrained species (e.g., Ca(BH$_4$)$_4^{2-}$) in the environment of the remaining unconstrained species. Since our system contains two calciums and four borohydrides, the unconstrained space is different depending on the value of the collective variables. In the Ca(BH$_4$)$_4^{2-}$ basin, the unconstrained Ca can only be fully solvated. On the other hand, in the Ca2+ basin, the other calcium can exist in five different ion coordination states (Ca$^{2+}$, Ca(BH$_4$)$^+$, Ca(BH$_4$)$_2$, Ca(BH$_4$)$_3^-$, and Ca(BH$_4$)$_4^{2-}$). This is the reason why no symmetry is expected on the free energy surface along the coordination axis, e.g., between CN = 0 and CN = 4 and CN = 1 and CN = 3.

## Population sampling

The structures and equilibrium population of points of interest in the free energy surface were collected using Umbrella Sampling. Initial structures were obtained from metadynamics trajectory snapshots at the desired point in the CV space. The collective variable coordinates

were restrained with a harmonic potential centered at the desired value, with force constant $1/w^2$ where $w$ is the width of the collective variable grid (see Table S1). Then, trajectories of 150–200 ns were calculated under similar MD parameters as the final equilibration step and MTD run.

## Data analysis

Umbrella sampling trajectories were analyzed with an unsupervised learning methodology recently developed by us[20]. In this protocol, between 5000 and 10,000 local atomic arrangements were extracted from each US trajectory and aligned while taking into account possible permutations between similar elements (e.g., THF - O)[59,60]. Classification based on their structural similarity was carried out using dimensionality reduction[61,62] and clustering[63] algorithms in an ASE-compatible[64] environment. The $n$-dimensional free energy landscapes obtained from the MTD sampling were explored with a Jupyter-adapted version of the MEPSAnd module[65] in order to find critical points and minimum energy pathways.

## Continuum model

The free energy of a system formed by a finite concentration of multiple (charged) species by a (charged) electrode was minimized using a beyond Gouy–Chapman model based on a modified Poisson–Boltzmann equation[40–42]. This continuum model accounts for the electrostatic potential of the electrode as well as each species' specific adsorption (the so-called Frumkin effects)[66–69], chemical potential and entropic (volume exclusion) effects, and chemical re-equilibration between species; parameterized from free-energy sampling results.

## Electron transfer rate estimates

The electron transfer rate estimate $I$ was obtained by integrating the individual contributions of isomers over the interface:

$$I \propto \sum_s \int_z dz\, C_s(z)\, e^{-z/z_0}\, e^{-\Delta G_s/k_B T} \tag{2}$$

where $C_s(z)$, each isomer's concentration profile, was obtained by renormalizing the species' concentration profiles of the continuum model with the US/UL isomer layer populations. $z_0$ is the electron tunneling decay length, set to 1 nm here. The reduction potential for a given isomer, $\Delta G_s$, was obtained using Density Functional Theory (see SI Section 2).

## Data availability

Additional computational details and free-energy information referenced in the text can be found in the Supporting Information. Source data for Figs. 1–6 are provided with this paper in the Supplementary Data file. The raw datasets generated and analyzed during the current study are also available from the corresponding author upon reasonable request. Source data are provided in this paper.

## Code availability

The continuum model code is available upon request to the corresponding author.

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

## Acknowledgements

This work was primarily supported by the Joint Center for Energy Storage Research (JCESR), an Energy Innovation Hub funded by the U.S. Department of Energy, Office of Science, Basic Energy Sciences. The theoretical analysis in this work by ASM was supported by a User Project at The Molecular Foundry and its computing resources, managed by the High-Performance Computing Services Group at Lawrence Berkeley National Laboratory (LBNL), supported by the Director, Office of Science, Office of Basic Energy Sciences, of the United States Department of Energy under Contract DE-AC02-05CH11231. We thank Dr. Artem Baskin and the Reviewers for inspiration.

## Author contributions

A.S.M. performed the calculations, analyzed the data, and wrote the original draft with support from D.P. A.S.M. and F.R. developed and tested the clustering algorithm. S.S. contributed to force field development. D.P. supervised and managed the project and developed and wrote the continuum model framework. All authors contributed to editing the paper.

## Competing interests

The authors declare no competing interests.
