## [Peer Review File · Nature Communications]

Ca-dimers, solvent layering, and dominant electrochemically active species in $\text{Ca}(\text{BH}_4)_2$ in THFREVIEWER COMMENTS

Reviewer #1 (Remarks to the Author):

This manuscript studied $\text{Ca}(\text{BH}_4)_2/\text{THF}$ electrolyte through molecular dynamic simulations. The solvated species were analyzed in detail and the main charged species were determined to be $\text{Ca}(\text{BH}_4)^+$ and $\text{Ca}(\text{BH}_4)_3^-$, while most salt existed as dimers or monomers. An in-depth examination of the electrode-electrolyte interface revealed a sharp increase in the concentration of $\text{Ca}(\text{BH}_4)^+$ species generated by the disproportionation of salt dimers when negative bias was applied. The study's findings provided insight into the molecular-scale behavior of the $\text{Ca}(\text{BH}_4)_2$ electrolyte and highlighted the importance of free energy sampling to attack such complex problems. This manuscript is interesting and suggested to be published after handling the following minor issues.

1. Given its high population in the vicinity of the electrode upon charging, CaBH_4^+ might be the strongest candidate for the electroactive species in this electrolyte. However, the reduction potential is also a crucial factor in determining the electroactive species. Therefore, the reduction potential of each species is suggested to be calculated.
2. Electric double layer has an important influence on the formation of SEI in batteries. At the biased interfaces, has the influence of the electric double layer been taken into account?
3. Based on the obtained electroactive species, what are the potential components of SEI?
4. The capitalization of the subheadings is inconsistent. "At the electrode-electrolyte interface", "Biased Interfaces"...
5. In Line 533, while the generation of electroactive species can be a plausible explanation for the initial overpotential spike in the stripping/plating profile, other factors like the passivating CaH_2 layer formed on Ca metal surface may also account for this phenomenon. Thus, this experimental observation may not be supportive enough evidence for the proposed mechanism.
6. The full name of "IDL" (intermediate density layer) appears multiple times throughout the manuscript. Please check the manuscript carefully. Typo to be corrected: Eg. Line 724 $\text{Ca}(\text{BH}_1)^+$

Reviewer #2 (Remarks to the Author):

This work studies the formation of electrochemically active species in $\text{Ca}(\text{BH}_4)_2$ in THF electrolytes through molecular dynamics simulations. Free-energy analysis indicates that this electrolyte has a majority population of neutral Ca dimers and monomers, albeit with diverse molecular conformations as revealed by unsupervised learning techniques but with an order of magnitude lower concentration of possibly electroactive charged species, such as the monocation, CaBH_4^+ . The dense layering of THF molecules within 1 nm of the electrode surface (modeled here using graphite) hinders the approach of reducible species to within 0.6 nm and instead enhances the local concentration of species in a narrow intermediate-density layer from 0.7-0.9 nm. Accordingly, the Ca-dimers and solvent layering structure

near the electrode were concluded to be two determining factors in the formation of active species. Generally, this work is well organized and affords a deep sight into the solvation structure in both bulk and EDL. The understanding of the formation of CaBH_4^+ active species delivers reveals the working mechanism of $\text{Ca}(\text{BH}_4)_2$ THF electrolytes. It is suggested to be published in Nature Communications after addressing the following issues.

1. The reason for focusing on $\text{Ca}(\text{BH}_4)_2$ THF electrolytes is not fully explained. It is far from enough to mention it with a Nature Materials paper published in 2018. What are the major advantages and remaining issues for the electrolytes? Is the formation of CaBH_4^+ active species in the electrolytes a critical point for the further study of the electrolytes?
2. Currently, a very detailed discussion of the simulation results is presented from the simulation side. However, the discussion on the underlying chemistry is a bit lacking, especially for the comparison with experimental results. It is highly suggested to enhance the discussion of the new science brought from the simulation results rather than mainly presenting the results.
3. Only two $\text{Ca}(\text{BH}_4)_2$ molecules were introduced into the simulations box. A low density of the species may impede adequate sampling even though an enhanced sampling method was used. It is suggested to demonstrate the convergence and reproducibility of the simulation results. Currently, not too much data are provided to support these.
4. Following the last question, the concentration of the modeled electrolytes is much lower than practical electrolytes. It is suggested to check the influence of the salt concentration, which has a significant influence on the solvation structure and transport mechanism. CaBH_4^+ active species may be not necessary in highly concentrated conditions.
5. Two specific charge states with (1) 0.065 and (2) 0.13 e/nm² were considered in the case of biased interfaces. It is suggested to provide the corresponding electrode potential with the electrode charge density. The electrode potential is of more practical meaning than the Fermi level.
6. Molecular dynamics simulation is a powerful tool for probing electrolyte structure and physicochemical properties. It is suggested to refer to recent review papers, such as Chem. Rev. 2022, 122, 10970–11021 and Chem. Rev. 2019, 119, 7940–7995. Besides, the ACR paper may be helpful to understand the structure–function relationship of electrolytes (Acc. Chem. Res. 2020, 53, 1992–2002).

Reviewer #3 (Remarks to the Author):

In this work, the authors explore the formation of electroactive species in the electrolyte $\text{Ca}(\text{BH}_4)_2$ in THF through molecular dynamics simulation. It is found that this electrolyte has a majority population of neutral Ca dimers and monomers, meanwhile, the monocation, CaBH_4^+ would be the possibly electroactive charged species. The dense layering of THF molecules enhances the local concentration of species in a narrow intermediate-density layer. Based on the provided results and discussions, this work can be recommended to publish in this journal after major revision. The specific comments are indicated below:

1. According to the results of the free-energy surface sampled using metadynamics with respect to Ca-Ca distance and Ca-B(H₄) coordination number in Fig. 1c, various neutral monomers would be appeared. Therefore, the high-resolution mass spectrometry is suggested to carry out with the aim to determine the main complexes.
2. The X-ray photoelectron spectroscopy is suggested to employ, which can facilitate to understand the charged species at the interface of electrolyte.
3. The galvanostatic charge/discharge profiles for plating and stripping of Ca are suggested to provide, which would support the understanding the evolution of the THF solvent layering after the first cycle. The corresponding results and discussions on the reductive instability of

the coordinating
species are required to present.

Response to Reviewers: Ca-dimers, solvent layering, and
dominant electrochemically active species in $\text{Ca}(\text{BH}_4)_2$ in THF

January 5, 2024

Reviewer #1 (Remarks to the Author)

This manuscript studied $\text{Ca}(\text{BH}_4)_2/\text{THF}$ electrolyte through molecular dynamic simulations. The solvated species were analyzed in detail and the main charged species were determined to be $\text{Ca}(\text{BH}_4)^+$ and $\text{Ca}(\text{BH}_4)_3^-$, while most salt existed as dimers or monomers. An in-depth examination of the electrode-electrolyte interface revealed a sharp increase in the concentration of $\text{Ca}(\text{BH}_4)^+$ species generated by the disproportionation of salt dimers when negative bias was applied. The study's findings provided insight into the molecular-scale behavior of the $\text{Ca}(\text{BH}_4)_2$ electrolyte and highlighted the importance of free energy sampling to attack such complex problems. This manuscript is interesting and suggested to be published after handling the following minor issues.

We thank the Reviewer for their positive assessment of our work and appreciation of its value and relevance to the field.

Comment 1

Given its high population in the vicinity of the electrode upon charging, CaBH_4^+ might be the strongest candidate for the electroactive species in this electrolyte. However, the reduction potential is also a crucial factor in determining the electroactive species. Therefore, the reduction potential of each species is suggested to be calculated.

Agreed. We have now estimated the reduction potential of each species in the bulk electrolyte and included its modulation with distance from the electrode. The combination of concentration profiles and reduction potentials with a phenomenological estimate of the electron tunneling length into the electrolyte provides us with an effective electron transfer rate that we have computed and summarized in **(new) Figure 6**. The major conclusion of this addition is that it confirms our initial suspicion that the dominant electroactive species is the monocation. It also indicates that electron transfer in this electrolyte should be dominated by inner-sphere processes, i.e., electron transfer to monocations quite close to the interface with necessarily reduced solvent coordination. Surprisingly, we observed that some nanoscale separation between electroactive monocations and the electrode surface (perhaps due to the presence of the SEI, which we model here as a dielectric spacer) can further enhance the electron transfer rate and effectively reduce the overpotential - in agreement with recent observations for this electrolyte that the overpotential reduces from 250 mV to 100 mV after the first charge.⁽¹⁾

Admittedly, these are only preliminary steps to computing actual electron transfer rates and still a lot of work is needed to determine the parameters necessary for the application of Marcus-Hush-Chidsey approaches to the study of complex electrolytes such as these.

After deeper consideration of reductive stability, we have made the following changes (**highlighted in yellow**) in the manuscript:

- **added a new section** in the main text titled '**Reductive processes and the seeds of**

the SEI' (lines 528 - 639).

- **added Figure 6.**
- **updated** the **Discussion** section in light of these new results (lines 684 - 692).
- **edited** the **Methods** section accordingly, to include a description of the methodology used to calculate the effective electron transfer rate (lines 909 - 918).
- **added** computational details regarding the calculation of the **reduction potentials** in the **Supporting Information** (SI Sections 2 and 3)
- **added Figure S7** in the Supporting Information, showing concentration effects in the effective electron transfer rate.

Comment 2

Electric double layer has an important influence on the formation of SEI in batteries. At the biased interfaces, has the influence of the electric double layer been taken into account?

In the previous version of the manuscript, the proper account for the presence of the electric double layer was necessarily limited by the finite size of our molecular dynamics simulations. However, now we have augmented our study with the inclusion of a continuum model that provides modified Poisson-Boltzmann estimates of the population of each relevant species (dication, monocation, complex anion, borohydride anion, neutral monomer and neutral dimer) with respect to variations in concentration and electrode potential. This takes into account not only EDL effects, but also the role of the adsorption free energy profile for each species – so-called Frumkin effects.^(2,3,4,5)

There are multiple effects of the EDL on the final species profiles near the electrode, which are summarized as **updates to Figures 3 b and 4 b**. We replaced our old population estimates derived purely from MD with the continuum model profiles. These indicate clearly the “activation” of the interfacial region with charged species due to negative potentials and the associated competition for space with the neutral species which are attracted to the intermediate and dense solvent layers.

These effects are now incorporated within **Figure 6** in terms of influencing the calculation of the reduction potential of each species as a function of distance from the electrode. The electric potential stabilizes charged species, which increases the magnitude of their required reduction potential. The local concentrations of each species also update the reduction potential via the concentration (activity) dependence of our free energy - equivalent to the Nernst equation. And, since the local concentrations are computed self-consistently via the Poisson-Boltzmann equation, we believe we are now accounting for double layer effects in our identification of the primary electroactive species - the monocation, CaBH_4^+ .

We imagine that reduction of this monocation species, and associated decomposition of nearby coordinating solvent molecules is the primary source of SEI formation, leading to both organic and boroxide components. We cannot speculate further on subsequent products, since our study only provides us access to the initial reductive steps at the electrode. However, we have

provided a key step for further studies on this topic - the identification of the undercoordinated monocation in the dense layer as the key species whose reduction contributes to the observed current just above the thermodynamic reduction potential for Ca deposition.

Besides the changes already mentioned in the response to the previous comment, we have:

- **updated Figures 3 b and 4 b**
- **updated the main text** results regarding the EDL at the neutral (lines 283 - 297, 332 - 345) and negatively charged interface (lines 377 - 397 and 427 - 433) in light of the continuum model results shown in Fig. 3 b and Fig. 4 b.
- **added** a description of the continuum model in the **Methods** Section (lines 989 - 908)
- **edited the Discussion** section to include EDL effects and the ‘activation’ of the electrolyte (lines 655 - 673) as well as our insight into possible sources of SEI formation (701 - 707).
- **added Figure S6** in the Supporting Information showing the populations of monocation and dication isomers at the charged interface including a full solvation sphere (i.e. complete solvent molecules).

Comment 3

Based on the obtained electroactive species, what are the potential components of SEI?

As stated above, since the primary reduced species is the undercoordinated monocation – $[\text{CaBH}_4^+](\text{THF})_4$ – then we would expect, as proposed previously,⁽⁶⁾ that the components of the SEI would be oxidized boron and mixed organics, from decomposition of both anions and solvent molecules. In that previous work, we proposed a mechanism of anion and solvent decomposition that requires the coincidence of free borohydride anions with Ca-coordinated THF molecules, as polarization of the β -C-H bond renders coordinated THF susceptible to nucleophilic attack. With our gained knowledge of the interface, we can add more context to this mechanism. Following monocation reduction in the dense layer (DL), BH_4^- nucleophilic attack on a coordinating THF molecule of one of the abundant nearby DL dications (**see new Fig. 6**) may lead to decomposition of both species. Subsequent release of H_2 molecules which could further react with the reduced Ca to form hydride deposits in the anode (as observed by electron energy loss spectroscopy and cryo-electron microscopy). We can speculate that this reduction-triggered chemical decomposition of the solvent may be concentration dependent, as electrolytes with lower concentration require much higher overpotentials to obtain a current, which in turn increases the local concentration of ions at the interface and the probability of a decomposition event post reduction.

Comment 4

4. The capitalization of the subheadings is inconsistent. “At the electrode-electrolyte interface”, “Biased Interfaces”...

We thank the reviewer for the attention and have **edited subheadings** for consistency.

Comment 5

In Line 533, while the generation of electroactive species can be a plausible explanation for the initial overpotential spike in the stripping/plating profile, other factors like the passivating CaH_2 layer formed on Ca metal surface may also account for this phenomenon. Thus, this experimental observation may not be supportive enough evidence for the proposed mechanism.

In light of the results obtained with the continuum model, we have reconsidered our explanation of the overpotential profile. While a negative potential difference (at least -1.1 V at 1.65 M) is necessary to ‘activate’ the electrolyte and generate electroactive species (i.e. undercoordinated monocations, $[\text{CaBH}_4^+](\text{THF})_4$ in the DL), the magnitude of this potential is significantly smaller than the thermodynamic potential for Ca deposition (-2.25 V). At potential differences at or slightly larger than the deposition potential, the bare electrode can stabilize a significant population of dications (solvated Ca^{2+}) in the DL, which displaces the more easily reduced monocation population further from the electrode, thereby requiring some overpotential to generate a significant deposition current. We fixed on the observed 250 mV overpotential to define the associated significant electron transfer rate. However, to generate an equivalent current (electron transfer rate) the required overpotential is significantly reduced if the electrode is covered with a dielectric spacer (acting as a very simple model of an SEI) as shown in the new Figure 6 b. We suggest that the formation of a thin SEI layer may be the cause of the experimentally observed decrease in overpotential from 250 to 100 mV following first charge.

We agree with the reviewer that other phenomena may contribute to the thermodynamic overpotential. The formation of Au-Ca alloys^(7,6), the influence of NaBH_4 ‘impurities’⁽⁶⁾ or hydride transfer from borohydride⁽⁸⁾ have been suggested to ‘activate’ the surface and decrease nucleation overpotential. However, recent, hydride-sensitive chemical mapping of Ca deposits using EELS has shown that CaH_2 forms inclusions within the *bulk* of the anode and not on its surface,⁽⁶⁾ so we do not consider the hydride as a source of overpotential here.

We have updated the manuscript as follows:

- **edited** the **Discussion** section to reflect the new explanation (lines 685 - 701).
- **added** a description of our **new results** on overpotential as a function of dielectric spacer in Section ‘Reductive processes and the seeds of the SEI’ (lines 605 - 652).

Comment 6

The full name of “IDL” (intermediate density layer) appears multiple times throughout the manuscript. Please check the manuscript carefully. Typo to be corrected: Eg. Line 724 $\text{Ca}(\text{BH}_1)^+$

We have fixed the typo in (now) line 869.

Reviewer #2 (Remarks to the Author)

This work studies the formation of electrochemically active species in $\text{Ca}(\text{BH}_4)_2$ in THF electrolytes through molecular dynamics simulations. Free-energy analysis indicates that this electrolyte has a majority population of neutral Ca dimers and monomers, albeit with diverse molecular conformations as revealed by unsupervised learning techniques but with an order of magnitude lower concentration of possibly electroactive charged species, such as the monocation, CaBH_4^+ . The dense layering of THF molecules within 1 nm of the electrode surface (modeled here using graphite) hinders the approach of reducible species to within 0.6 nm and instead enhances the local concentration of species in a narrow intermediate-density layer from 0.7-0.9 nm. Accordingly, the Ca-dimers and solvent layering structure near the electrode were concluded to be two determining factors in the formation of active species. Generally, this work is well organized and affords a deep sight into the solvation structure in both bulk and EDL. The understanding of the formation of CaBH_4^+ active species delivers reveals the working mechanism of $\text{Ca}(\text{BH}_4)_2$ THF electrolytes. It is suggested to be published in Nature Communications after addressing the following issues.

We greatly appreciate the Reviewer's favorable evaluation of our work.

Comment 1

1. The reason for focusing on $\text{Ca}(\text{BH}_4)_2$ THF electrolytes is not fully explained. It is far from enough to mention it with a Nature Materials paper published in 2018. What are the major advantages and remaining issues for the electrolytes? Is the formation of CaBH_4^+ active species in the electrolytes a critical point for the further study of the electrolytes?

We have added to the introduction (lines 54-75) of the manuscript to provide additional references, a more complete review of the $\text{Ca}(\text{BH}_4)_2/\text{THF}$ field, and outlined the listed challenges. We have framed this information in the greater context of multivalent-ion electrolytes and their challenges. We would like to remark that (at least) four experimental studies have been published in this and related electrolytes since our initial submission, and we refer to them within our revised manuscript: Refs. 9,10,11,12. Changes have been highlighted in yellow in the manuscript.

Comment 2

Currently, a very detailed discussion of the simulation results is presented from the simulation side. However, the discussion on the underlying chemistry is a bit lacking, especially for the comparison with experimental results. It is highly suggested to enhance the discussion of the new science brought from the simulation results rather than mainly presenting the results.

We thank Reviewer 2 for this observation. To provide more details on the underlying chemistry, we have added (as stated above) a more thorough summary of the state of this field of research in the introduction; we have also included a descriptive chemical equation to highlight which species are present in the electrolyte (**Eq. 1**). With the inclusion of our new continuum model results, we now provide more experimentally accessible local concentrations of each species and combine this information with estimates of the reduction potential to provide an analog of the current (effective electron transfer rate) as a function of overpotential with respect to the thermodynamic potential for Ca deposition.

The insights from these additional results are now incorporated into our discussion at the end of the manuscript and connected directly to characterization of electrolytes and electrochemical observations of the performance of this system as a function of concentration and overpotential. Note that the only interfacially-sensitive measurements of this electrolyte that we are aware of have been recently reported by Yang *et al.*⁽⁹⁾ and McClary *et al.*,⁽⁶⁾ and focus on SEI composition. As far as we know, the only possible interfacially-sensitive measurement that could detect differences in the local coordination environment of calcium may be interfacial XAS. We intend to pursue this direction in future work, which would require to first assess if spectral differences between isomers would be large enough to differentiate between them once relative populations are taken into account.

Besides expanding the introduction and adding Eq. 1, we have:

- **updated Fig. 1c, 3b and 4b** to include species population profiles at finite concentrations accessible in experiment;
- obtained great agreement with experiment in the evolution of bulk species populations with concentration (**added** lines 177 - 182);
- provided a description of interfacial layering in terms of the Electric Double Layer (**added** lines 332 - 345);
- suggested a connection between the proposed mechanism of generation of active species with experimental performance studies (**added** lines 461 - 466);
- added electron-transfer results in (**new**) **Figure 6** which we discuss in (**new**) **section ‘Reductive processes and the seeds of the SEI’** (added lines 528 - 639);
- with the gained knowledge, **updated** the **Discussion** section on the experimental overpotential profile of this electrolyte (lines 485 - 493).

Comment 3

Only two $\text{Ca}(\text{BH}_4)_2$ molecules were introduced into the simulations box. A low density of the species may impede adequate sampling even though an enhanced sampling method was used. It is suggested to demonstrate the convergence and reproducibility of the simulation results. Currently, not too much data are provided to support these.

We agree with the reviewer and we thank them for the inspiration that this comment provided to incorporate a concentration- and potential-dependent continuum model in the new version of our manuscript. This afforded us the possibility to extend our (admittedly) limited molecular dynamics results to finite concentrations and experimentally relevant electrode potentials.

Key insights gained from this addition include: (1) The observation of a crossover in relative population of neutral dimers with respect to neutral monomers in the bulk electrolyte, in agreement with published observations;^(7,13) (2) The strong dependence of the interfacial population and reduction potential on concentration. Ultimately, we found orders of magnitude increase in the effective electron transfer rate upon increasing concentration from 0.12 M to 1.65 M, which matches experience in the literature^(7,9) with the performance of electrochemical cells based on this electrolyte.

We have included all of these details in the updated manuscript. Specifically, we have:

- **updated Fig. 1 c** to include species evolution with concentration in the bulk, and updated the text to discuss its agreement with experiment (lines 177 - 185).
- **updated Figures 3b and 4b** with finite concentration profiles of species at the neutral and negatively charged interface obtained with the continuum model.
- **updated the main text results section** on the neutral (lines 283 - 297, 332 - 345) and negatively charged interface (lines 377 - 397 and 427 - 433) in light of the updates to Fig. 3 b and Fig. 4 b.
- accounted for the distance dependence (with respect to the electrode surface) of: the adsorption potentials, local concentrations and overall electrostatic potential, which allows us to provide a detailed physical model for the reduction potential of each species as it approaches the electrode and to use this to estimate the effective electron transfer rate (summarized in new section: **Reductive processes and the seeds of the SEI** with details in the **Methods**).

Note that low concentration metadynamics (0.12 M) and continuum model results at the same concentration are in agreement.

Comment 4

Following the last question, the concentration of the modeled electrolytes is much lower than practical electrolytes. It is suggested to check the influence of the salt concentration, which has a significant influence on the solvation structure and transport mechanism. CaBH_4^+ active species may be not necessary in highly concentrated conditions.

As stated above, now that we have included concentration dependence through our continuum model, we can clearly see the influence of salt concentration on observed performance in electrochemical cells employing this electrolyte. That said, the bulk electrolyte is still practically devoid of charged species and therefore we still rely on our proposed disproportionation mechanism to generate charged species at the interface with the electrode.

Additionally, we find that the concentration of fully solvated dications is not increased at all with increasing concentration. And, in the limiting case where, at 1.65 M, reduction potentials do lead to a population of Ca^{2+} in the dense layer close to the bare electrode, we find that their specific reduction potential is too high for them to contribute to the current and, with the presence of any intervening dielectric layer (such as the SEI) that the corresponding potential difference might not be enough to support the presence of dications at all at the interface. Therefore, we most likely rely entirely on the monocation population for the initial current when plating and additionally rely on a disproportionation mechanism from nearby dimers to replenish it as it is consumed.

We have included these new results in the **new section** ‘*Reductive processes and the seeds of the SEI*’ (lines 528 - 639) and **Fig. 6**; and we have **updated** the **discussion** to include concentration effects on interfacial population of active species, their reduction and replenishment mechanism in lines 659 - 686 and 697 - 717.

Comment 5

Two specific charge states with (1) 0.065 and (2) 0.13 e/nm² were considered in the case of biased interfaces. It is suggested to provide the corresponding electrode potential with the electrode charge density. The electrode potential is of more practical meaning than the Fermi level.

We agree with the reviewer. Armed with a continuum model description of the electrolyte interface, we can now assign these charge states to specific potential differences and have done so in the manuscript. (These are potential differences with respect to the bulk electrolyte, which are not necessarily easy to measure - requiring a reference electrode.) Ultimately, we find that they represent quite small potential differences relative to the thermodynamic potential for deposition of Ca. As now stated (lines 366-373):

‘These surface charge densities correspond to potential differences of -0.26 V (CS1) and -0.42 V (CS2), calculated using our continuum model (see Methods) assuming a simulated bulk concentration of 0.12 M. (Note that at higher bulk concentrations, with a shorter Debye screening length, this estimated potential difference should be even smaller – at 1.65 M: -0.20 V (CS1) and -0.35 V (CS2).)’

Additionally, we have **converted** our charge densities to **potential differences** throughout the manuscript to facilitate comparison with experiment.

Comment 6

Molecular dynamics simulation is a powerful tool for probing electrolyte structure and physicochemical properties. It is suggested to refer to recent review papers, such as Chem. Rev. 2022, 122, 10970–11021 and Chem. Rev. 2019, 119, 7940–7995. Besides, the ACR paper may be helpful to understand the structure–function relationship of electrolytes (Acc. Chem. Res. 2020, 53, 1992–2002).

We fully agree with the reviewer in that molecular dynamics simulations, especially when combined with enhanced sampling and continuum models, can provide great insight into electrolyte structure and properties. We thank them for the suggested review papers. We have included the citations:

- Chem. Rev. 2022,122, 10970–11021 and Acc. Chem. Res. 2020, 53, 1992–2002:
‘Molecular dynamics provides a window to the inner workings of electrolytes, revealing details of coordination of cations by solvent molecules and anions.^(14,15)’ (Line 39)
- Chem. Rev. 2019, 119, 7940–7995:
Since we do not employ polarizable force-fields in this study, we included this citation as follows (line 735 - 738).
‘Accurate inclusion of Debye screening from finite ionic concentrations, and dielectric screening due to solvent or anion polarizability⁽¹⁶⁾ (which we approximate only by charge rescaling) should be considered in future studies.’

Reviewer #3 (Remarks to the Author)

In this work, the authors explore the formation of electroactive species in the electrolyte $\text{Ca}(\text{BH}_4)_2$ in THF through molecular dynamics simulation. It is found that this electrolyte has a majority population of neutral Ca dimers and monomers, meanwhile, the monocation, CaBH_4^+ would be the possibly electroactive charged species. The dense layering of THF molecules enhances the local concentration of species in a narrow intermediate-density layer. Based on the provided results and discussions, this work can be recommended to publish in this journal after major revision. The specific comments are indicated below:

We thank the Reviewer for their careful consideration of our manuscript and their positive evaluation.

Comment 1

According to the results of the free-energy surface sampled using metadynamics with respect to Ca-Ca distance and Ca-B(H_4) coordination number in Fig. 1c, various neutral monomers would be appeared. Therefore, the high-resolution mass spectrometry is suggested to carry out with the aim to determine the main complexes.

As the reviewer points out, our analysis of the bulk populations indicates that neutral monomers $\text{Ca}(\text{BH}_4)_2$ may not only have different formulas depending on their degree of solvation (i.e. 3 or 4 THF molecules) but also different stereoisomers for each formula (Fig. 1 in the manuscript).

Unfortunately, in mass-spectrometry, the time elapsed during the ionization process and transit to the detector would allow for re-equilibration of species by the time they reach the detector. Mass-spectrometry may only provide the chemical formulae of different species, but not the possible conformations and configurations, which is part of the insight that we provide in our analysis. Multimer neutral species with many conformations and configurations, such as dimers, would present a similar challenge, especially given the small barriers for interconversion of species (e.g., dimer dissociation or disproportionation).

In Ref. 8, Ta *et al.* performed liquid injection field desorption ionization mass spectrometry (LIFDI-MS) and electrospray ionization mass spectrometry (EI-MS) on cycled $\text{Ca}(\text{BH}_4)_2/\text{THF}$ electrolyte solutions in positive ion mode. The LIFDI-MS results were interpreted as only containing $\text{BH}_x \cdot \text{THF}$ complexes. EI-MS results were interpreted as different Ca-THF complexes and fragments, up to $[\text{Ca}(\text{THF})_{10}(\text{CH}_7\text{O})]^+$ ions, none of them seemingly containing BH_4^- . According to our simulations and previous reports^(7,13,17,18) those complexes may not be representative of the calcium solvation environment in the liquid, room-temperature electrolyte.

On the other hand, measurements aimed at determining the main complexes and their relative populations in the bulk have been reported recently using DRS⁽¹³⁾ and EXAFS⁽⁷⁾. Our **new results** using a continuum model are in great agreement with their reported evolution of bulk

populations of dimers and neutral species with concentration. We have **updated Fig. 1 c** and the **Methods** section, and **added** lines 177 - 185 to reflect this. Changes have been **highlighted in yellow** in the manuscript.

We also obtain good agreement in terms of interatomic distances in dimers (lines 222 - 225).

Comment 2

The X-ray photoelectron spectroscopy is suggested to employ, which can facilitate to understand the charged species at the interface of electrolyte.

Interfacial AP-XPS experiments are very difficult to realize - particularly for a more volatile liquid such as THF (vs. water) - and would require a high background pressure in the chamber to maintain a liquid phase at the interface. This, in turn, could degrade the measured surface signal, as electrons have to pass through this denser medium. In addition, such measurements rely heavily on theoretical interpretation. In our work, we are showing that even if such methods can be applied there is a wealth of species that would have to be considered in the analysis - each species' XPS signal would need to be sufficiently different; taking into account individual populations and their change depending on concentration and surface potential.

So far, the only interfacially-sensitive measurements of this electrolyte that we are aware of have been recently reported by Yang *et al.*⁽⁹⁾ and McClary *et al.*,⁽⁶⁾ and focus on SEI composition. The TEM and EELS cryo-measurements investigating the composition of the solid electrolyte interphase were later confirmed by Yang *et al.*⁽⁹⁾ using XPS. In addition, they performed electrochemical quartz crystal microbalance with dissipation (EQCM-D) measurements on the electrolyte interface during cycling. Their results were interpreted as indicating a high monocation population at the interface. The stark changes in mass they see prior to plating, at 0-2 $\mu\text{C}/\text{cm}^2$ surface charge,⁽⁹⁾ are compatible with our free-energy and new continuum model results that show a rapid change in interfacial composition with increased negative electrode potential differences.

Comment 3

The galvanostatic charge/discharge profiles for plating and stripping of Ca are suggested to provide, which would support the understanding the evolution of the THF solvent layering after the first cycle. The corresponding results and discussions on the reductive instability of the coordinating species are required to present.

We would like to note that cyclic voltametry profiles and galvanostatic charge-discharge diagrams of calcium plating/stripping for this electrolyte have been reported previously by Wang *et al.*⁽¹⁾, Hahn *et al.*⁽⁷⁾ and Yang *et al.*⁽⁹⁾, the latter two also including concentration-dependence, and by McClary *et al.*,⁽⁶⁾ with the observations for this electrolyte that the overpotential reduces from 250 mV to 100 mV after the first charge.⁽¹⁾

Our free energy calculations indicate that solvent layering changes little with increasing potential differences (Supporting Information, Fig. S4). Thanks to our new continuum model results, we

have investigated the evolution of species at the interface at pre-deposition negative biases, and **updated Figure 4 b** and its discussion (lines 382 - 395).

Additionally, we have now estimated the reduction potential of each species in the bulk electrolyte and included its modulation with distance from the electrode. The combination of concentration profiles and reduction potentials with a phenomenological estimate of the electron tunneling length into the electrolyte provides us with an effective electron transfer rate that we have computed and summarized in **(new) Figure 6**. The major conclusion of this addition is that it confirms our initial suspicion that the dominant electroactive species is the monocation. It also indicates that electron transfer in this electrolyte should be dominated by inner-sphere processes, i.e., electron transfer to monocations quite close to the interface with necessarily reduced solvent coordination. Surprisingly, we observed that some nanoscale separation between electroactive monocations and the electrode surface (perhaps due to the presence of the SEI, which we model here as a dielectric spacer) can further enhance the electron transfer rate and effectively reduce the overpotential - in agreement with the above-mentioned overpotential profile reported in experiments.⁽¹⁾

Since we agree with the reviewer on the importance of connecting the results of our simulations with relevant experimental evidence, a concern also brought up by other Reviewers, we have edited the manuscript as follows:

- we have placed our results in the context of existing experimental evidence by **expanding the introduction** with a more complete review of the $\text{Ca}(\text{BH}_4)_2/\text{THF}$ field
- **updated Fig. 1c, 3b and 4b** to include species population profiles at finite concentrations accessible in experiment;
- obtained great agreement with experiment in the evolution of bulk species populations with concentration (**added** lines 177 - 182);
- suggested a connection between the proposed mechanism of generation of active species with experimental performance studies (**added** lines 461 - 466);
- added electron-transfer results in **(new) Figure 6** which we discuss in **(new) section ‘Reductive processes and the seeds of the SEI’** (added lines 528 - 639);
- with the gained knowledge, **updated** the **Discussion** section on the experimental overpotential profile of this electrolyte (lines 485 - 493).

Finally, as far as we know, the only possible interfacially-sensitive measurement that could detect differences in the local coordination environment of calcium may be interfacial XAS. We intend to pursue this direction in future work, which would require to first assess if spectral differences between isomers would be large enough to differentiate between them once relative populations are taken into account.

References

- [1] Da Wang, Xiangwen Gao, Yuhui Chen, Liyu Jin, Christian Kuss, and Peter G. Bruce. Plating and stripping calcium in an organic electrolyte. *Nature Materials*, 17(1):16–20, 2018.
- [2] Allen J. Bard, Larry R. Faulkner, and Henry S. White. *Electrochemical methods : fundamentals and applications*. John Wiley & Sons, Ltd. Hoboken, NJ, Hoboken, NJ, 2022.
- [3] Lixin Fan, Yuwen Liu, Jiewen Xiong, Henry S. White, and Shengli Chen. Electron-transfer kinetics and electric double layer effects in nanometer-wide thin-layer cells. *ACS Nano*, 8(10):10426–10436, 2014. PMID: 25211307.
- [4] Aditya M. Limaye and Adam P. Willard. Modeling interfacial electron transfer in the double layer: The interplay between electrode coupling and electrostatic driving. *The Journal of Physical Chemistry C*, 124(2):1352–1361, 2020.
- [5] Andrew D. Pendergast and Henry S. White. Double-layer inhibition of peroxydisulfate reduction at mercury ultramicroelectrodes. a quantitative analysis of the frumkin effect including molecular transport and long-range electron transfer. *The Journal of Physical Chemistry C*, 127(23):11283–11297, 2023.
- [6] Scott A. McClary, Daniel M. Long, Ana Sanz-Matias, Paul G. Kotula, David Prendergast, Katherine L. Jungjohann, and Kevin R. Zavadil. A heterogeneous oxide enables reversible calcium electrodeposition for a calcium battery. *ACS Energy Letters*, 7(8):2792–2800, 2022.
- [7] Nathan T. Hahn, Julian Self, Trevor J. Seguin, Darren M. Driscoll, Mark A. Rodriguez, Mahalingam Balasubramanian, Kristin A. Persson, and Kevin R. Zavadil. The critical role of configurational flexibility in facilitating reversible reactive metal deposition from borohydride solutions. *J. Mater. Chem. A*, 8:7235–7244, 2020.
- [8] Kim Ta, Ruixian Zhang, Minjeong Shin, Ryan T. Rooney, Elizabeth K. Neumann, and Andrew A. Gewirth. Understanding ca electrodeposition and speciation processes in nonaqueous electrolytes for next-generation ca-ion batteries. *ACS Applied Materials & Interfaces*, 11(24):21536–21542, 2019. PMID: 31117456.
- [9] Zhenzhen Yang, Noel J. Leon, Chen Liao, Brian J. Ingram, and Lynn Trahey. Effect of salt concentration on the interfacial solvation structure and early stage of solid–electrolyte interphase formation in ca(bh₄)₂/thf for ca batteries. *ACS Applied Materials & Interfaces*, 15(20):25018–25028, 2023. PMID: 37171170.
- [10] Aaron M. Melemed, Dhyllan A. Skiba, Katherine J. Steinberg, Kyeong-Ho Kim, and Betar M. Gallant. Impact of differential ca²⁺ coordination in borohydride-based electrolyte blends on calcium electrochemistry and sei formation. *The Journal of Physical Chemistry C*, 0(0):null, 0.
- [11] Kazuaki Kisu, Rana Mohtadi, and Shin-ichi Orimo. Calcium metal batteries with long cycle life using a hydride-based electrolyte and copper sulfide electrode. *Advanced Science*, n/a(n/a):2301178.
- [12] Alan T. Landers, Julian Self, Scott A. McClary, Keith J. Fritzsche, Kristin A. Persson, Nathan T. Hahn, and Kevin R. Zavadil. Calcium cosalt addition to alter the cation solvation structure and enhance the ca metal anode performance. *The Journal of Physical Chemistry C*, 127(49):23664–23674, 2023.

- [13] Nathan T. Hahn, Julian Self, Kee Sung Han, Vijayakumar Murugesan, Karl T. Mueller, Kristin A. Persson, and Kevin R. Zavadil. Quantifying species populations in multivalent borohydride electrolytes. *The Journal of Physical Chemistry B*, 125(14):3644–3652, Apr 2021. doi: 10.1021/acs.jpcc.1c00263.
- [14] Nan Yao, Xiang Chen, Zhong-Heng Fu, and Qiang Zhang. Applying classical, ab initio, and machine-learning molecular dynamics simulations to the liquid electrolyte for rechargeable batteries. *Chemical Reviews*, 122(12):10970–11021, 2022. PMID: 35576674.
- [15] Xiang Chen and Qiang Zhang. Atomic insights into the fundamental interactions in lithium battery electrolytes. *Accounts of Chemical Research*, 53(9):1992–2002, 2020. PMID: 32883067.
- [16] Dmitry Bedrov, Jean-Philip Piquemal, Oleg Borodin, Alexander D. Jr. MacKerell, Benoît Roux, and Christian Schröder. Molecular dynamics simulations of ionic liquids and electrolytes using polarizable force fields. *Chemical Reviews*, 119(13):7940–7995, 2019. PMID: 31141351.
- [17] Diana Liepinya and Manuel Smeu. A computational comparison of ether and ester electrolyte stability on a ca metal anode. *Energy Material Advances*, 2021:9769347, 2021.
- [18] Yulin Jie, Yunshu Tan, Linmei Li, Yehu Han, Shutao Xu, Zhenchao Zhao, Ruiguo Cao, Xiaodi Ren, Fanyang Huang, Zhanwu Lei, Guohua Tao, Genqiang Zhang, and Shuhong Jiao. Electrolyte solvation manipulation enables unprecedented room-temperature calcium-metal batteries. *Angewandte Chemie International Edition*, 59(31):12689–12693, 2020.

REVIEWERS' COMMENTS

Reviewer #1 (Remarks to the Author):

The authors have well addressed my comments. I have no more questions.

Reviewer #2 (Remarks to the Author):

The manuscript has been well-revised and can be accepted now.

Reviewer #3 (Remarks to the Author):

Authors have revised the manuscript point by point. I recommend that the present version be accepted for publication.